# Navigating the garden of forking paths for data exclusions in fear conditioning research

Tina B Lonsdorf[1]*, Maren Klingelhöfer-Jens[1], Marta Andreatta[2,3], Tom Beckers[4], Anastasia Chalkia[4], Anna Gerlicher[5], Valerie L Jentsch[6], Shira Meir Drexler[6], Gaetan Mertens[7], Jan Richter[8], Rachel Sjouwerman[1], Julia Wendt[9], Christian J Merz[6]

[1]Department of Systems Neuroscience, University Medical Center Hamburg Eppendorf, Hamburg, Germany; [2]Department of Psychology, Biological Psychology, Clinical Psychology and Psychotherapy, University of Würzburg, Würzburg, Germany; [3]Instutute of Psychology, Education & Child Studies, Erasmus University Rotterdam, Rotterdam, Netherlands; [4]Centre for the Psychology of Learning and Experimental Psychopathology and Leuven Brain Institute, KU Leuven, Leuven, Belgium; [5]Faculty of Social and Behavioural Sciences, Programme group Clinical Psychology, University of Amsterdam, Amsterdam, Netherlands; [6]Institute of Cognitive Neuroscience, Department of Cognitive Psychology, Ruhr University Bochum, Bochum, Germany; [7]Department of Psychology, Utrecht University, Utrecht, Netherlands; [8]Department of Physiological and Clinical Psychology/ Psychotherapy, University of Greifswald, Greifswald, Germany; [9]Biological Psychology and Affective Science, University of Potsdam, Potsdam, Germany

*For correspondence:
t.lonsdorf@uke.de

**Abstract** In this report, we illustrate the considerable impact of researcher degrees of freedom with respect to exclusion of participants in paradigms with a learning element. We illustrate this empirically through case examples from human fear conditioning research, in which the exclusion of 'non-learners' and 'non-responders' is common – despite a lack of consensus on how to define these groups. We illustrate the substantial heterogeneity in exclusion criteria identified in a systematic literature search and highlight the potential problems and pitfalls of different definitions through case examples based on re-analyses of existing data sets. On the basis of these studies, we propose a consensus on evidence-based rather than idiosyncratic criteria, including clear guidelines on reporting details. Taken together, we illustrate how flexibility in data collection and analysis can be avoided, which will benefit the robustness and replicability of research findings and can be expected to be applicable to other fields of research that involve a learning element.

## Introduction

In the past decade, efforts to understand the impact of undisclosed flexibility in data collection and analysis on research findings have gained momentum – for instance in defining and excluding 'outliers' (*Simmons et al., 2011*). This flexibility has been referred to as 'researcher degrees of freedom' (*Simmons et al., 2011*) or 'the garden of forking paths' (*Gelman and Loken, 2013*) to reflect the fact that each decision during data processing and/or analysis will take the researcher down a different 'path'. Importantly and concerningly, these different paths can lead to fundamentally different end-points (i.e., results and associated conclusions) despite an identical starting point (i.e., raw data) (*Silberzahn et al., 2018*). Often, researchers take a certain path without malicious intent to obtain

favorable results (e.g., 'p-hacking'; *Head et al., 2015*): the decision to follow a certain path may be based on unawareness of alternative paths (due to lack of specific background knowledge) or the researcher following the most obvious path from an individual perspective. The latter is influenced by the scientific environment, the research question at stake or practices previously published by researchers in the field.

Admittedly, there is substantial ambiguity in what constitutes 'the best decision' for data analysis, and none of the available options may be necessarily incorrect (*Simmons et al., 2011*; *Silberzahn et al., 2018*). More precisely, different paths in the garden of forking paths may be more or less appropriate for different research questions, experimental designs, outcome measures or samples. Consequently, it is notoriously difficult for researchers, particularly those new to a field, to make informed and hence appropriate decisions. As a matter of fact, it is difficult to anticipate the number of different paths available and the consequences of choosing one over the other, or to come up with facts that truly justify choosing one path over the other – even for experts in a field. However, simply choosing a particular path because others chose it before (i.e., adopting published exclusion criteria) can also be highly problematic, as decisions often hinge on study-specific characteristics that do not invariantly apply to other studies.

We argue that it is important to raise awareness to this issue. Specifically, we think that it is critical to discuss both the rationale behind and the consequences associated with taking different analytical paths in general and in specific sub-fields of research. Here, we exemplarily take up this discussion for human fear conditioning research as a case example for tasks with a learning element grounded in recent discussions in science in general (*Flake and Fried, 2019*; *John et al., 2012*) and in fear conditioning research specifically (*Lonsdorf et al., 2017*; *Lonsdorf et al., 2019*). Fear conditioning is a typical paradigm employed to study (emotional) learning and memory processes with a particularly strong translational perspective (*Lonsdorf et al., 2017*; *Vervliet et al., 2013*). Questions addressed in the field of human fear conditioning are often concerned with consolidation, retrieval, generalization or modification of conditioned responses. Hence, it has often been claimed that the study of these processes requires the acquisition of a robust conditioned response as a precondition. Therefore, participants are often (routinely) excluded from analyses if they appear to not have learned ('non-learners') or not have been responsive to the experimental stimuli ('non-responders') during fear acquisition training, in which one conditional stimulus (CS+) predicts an upcoming aversive unconditioned stimulus (US) and another conditional stimulus does not (CS–) (*Lonsdorf et al., 2017*; *Pavlov, 1927*).

Critically, 'non-learning' is most often defined as a failure to show discrimination between the CS + and CS– in skin conductance responses (SCRs) – the most common outcome measure in the field (*Lonsdorf et al., 2017*). This practice may seem trivial at first glance and has been referred to as exclusion of 'non-learners', 'performance-based exclusion' or even 'exclusion of outliers'. Yet, defining a set of characteristics to identify individuals who 'did not learn' is operationalized in very heterogeneous ways across studies. The same applies to the criteria that determine what constitutes a' non-responder' during fear acquisition training.

In addition to the heterogeneity in operationalization, other problems of performance-based exclusion of participants are worth noting: definitions of 'non-learners' are typically based on SCRs only (for exceptions see *Ahmed and Lovibond, 2019*; *Oyarzún et al., 2019*) and 'non-learners' are typically excluded from *all* analyses, that is, all experimental phases and outcome measures of a study. As SCRs are not a pure measure of either learning or fear, but rather reflect arousal levels (*Hamm et al., 1993*) that serve as proxies for fear learning, classification into 'learners' and 'non-learners' on the basis of this single outcome measure may induce substantial sample bias. First, defining 'non-learning' on one single outcome measure, such as SCRs, ignores the fact that successful CS+/CS– differentiation may be present in other outcome measures (*Hamm et al., 1993*) such as fear potentiated startle (FPS) or ratings of fear and contingencies (i.e., cognitive awareness of the CS +/US contingencies). As such, 'non-learning' as defined on a single outcome measure such as SCRs cannot comprehensively capture 'non-learning'. Second, the level of responding in SCRs and CS+/ CS– discrimination has been shown to be associated with a vast number of individual difference factors (*Lonsdorf and Merz, 2017*; *Boucsein et al., 2012*) such as age and sex (for a discussion see *Boucsein et al., 2012*), ethnicity (*Alexandra Kredlow et al., 2017*; *Boucsein et al., 2012*), genetic make-up (*Garpenstrand et al., 2001*), use of oral contraceptives (*Merz et al., 2018b*) or personality traits (*Naveteur and Freixa I Baque, 1987*). Consequently, excluding participants from an

experiment as 'non-learners' may pre-select specific sub-samples and thus may thus severely hamper the generalizability and interpretation of the findings. Importantly, this practice may be a threat to and a limitation of the clinical translation of findings because it potentially leads to the selective exclusion of specific and highly relevant sub-groups. In fact, a recent meta-analysis suggests that patients suffering from anxiety disorders show overgeneralization of fear responding, which is enhanced when responding to the CS– (*Duits et al., 2015*), which may lead to reduced CS+/CS– discrimination if the response to the CS+ is comparable.

The concerns discussed above are merely based on theoretical considerations. Below, we aim to address the important and controversial topic of exclusion of 'non-learners' and 'non-responders' in human fear conditioning research empirically. We set out to provide an overview and inventory of the exclusion criteria that are currently employed in the field by means of a systematic literature search following PRISMA guidelines (*Moher et al., 2009*), covering a publication period of six months. Importantly, we distinguish between 'non-learners' (based on task performance, that is, CS+/CS– discrimination) and 'non-responders' (based on a lack of responsiveness) as assessed using SCRs. We expect the identified criteria for 'non-learners' and 'non-responders' to be characterized by noticeable heterogeneity (thus allowing for considerable researcher degrees of freedom) across studies. We thus aim to (1) raise awareness and (2) illustrate the impact of applying different exclusion criteria features (i.e., forking paths) on results and interpretation through case examples exemplified by the re-analyses of existing data sets. Finally, we aim to (3) provide

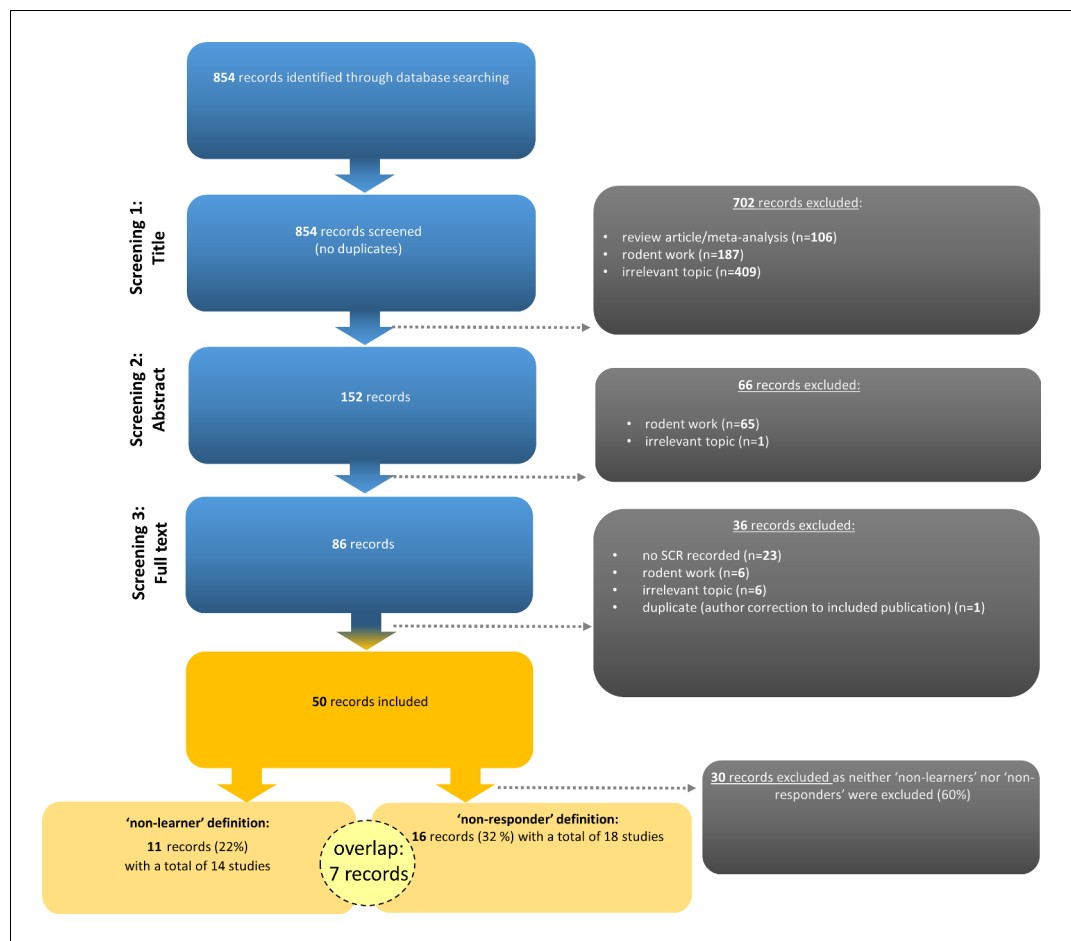

**Figure 1.** Flow chart illustrating the selection of records according to PRISMA guidelines (*Moher et al., 2009*). Note that seven records (14%) employed the definition and exclusion of both 'non-learners' and 'non-responders'. Examples of irrelevant topics included studies that did not use fear conditioning paradigms (see https://osf.io/uxdhk/ for a documentation of excluded publications).

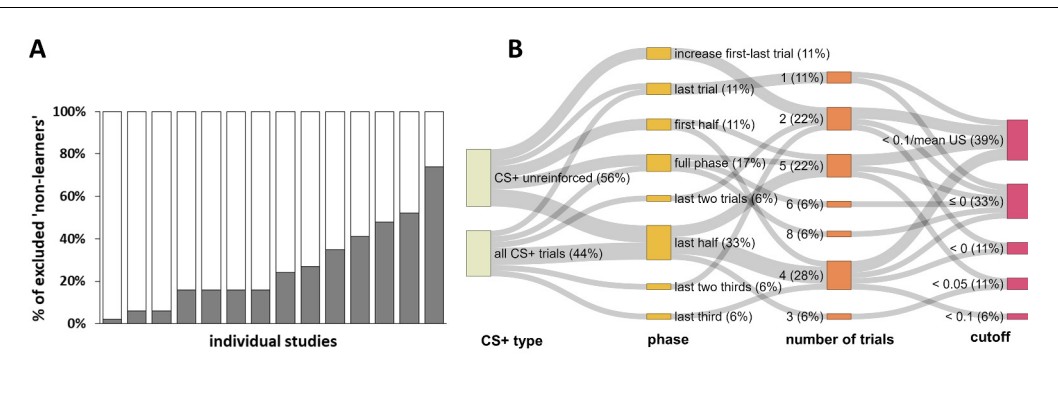

**Figure 2.** Graphical illustration of the percentage of 'non-learners' and forking path analysis across studies. (**A**) Illustration of the percentage of participants excluded ('non-learners') based on SCR CS+/CS–discrimination scores across studies included in the systematic literature search (note that these 14 individual studies are derived from 11 different records, as three records reported two individual studies each). Please note that some studies excluded participants on the basis of 'non-learning' as well as 'non-responding' (cf. *Figure 1*), and hence the percentages displayed here do not necessarily map onto the percentage of total participants excluded per study. Also note that the study with the highest percentage of excluded participants (i.e., 74%) reported the percentage of excluded participants as a single value that included 'non-learners' and 'non-responders'. This study is only included here because the largest proportion of exclusions can be expected to result from 'non-learning'. (**B**) Sanky plot showing the 'forking paths' of performance-based exclusion of participants as 'non-learners', illustrating differences in the experimental phase, number of trials, the SCR CS+/CS– discrimination score in μS used to define a 'non-learner', the CS+ type considered (illustrated as the nodes in graded colors) and their combinations used to define 'non-learners' across studies. Path width was scaled in relation to frequency of the combinations. Note that for some 'nodes' the percentages do not add up to 100% because of rounding.

methodologically informed, evidence-based recommendations for future studies with respect to defining and handling 'non-learners' and 'non-responders'.

## Results

### Definition of performance-based exclusion of participants ('non-learners') and number of participants excluded across studies

Slightly fewer than one fourth of the records (i.e., 22%; 11 out of 50 records comprising 14 individual studies as three records reported two studies each) included in the systematic literature search employed performance-based exclusion of participants (i.e., SCR 'non-learners', *Figure 1*).

Strikingly, every single one of these records used an idiosyncratic definition to define 'non-learners', yielding a total of eleven different definitions in the short period of six months (see *Appendix 1—table 1*). The percentages of excluded participants varied from 2% to 74% (*Figure 2A*) of the respective study sample. Definitions differed in i) the experimental (sub-)phases to which they were applied (i.e., whether the full phase or only the first half, second half or even single trials were considered), ii) the number of trials that the exclusion was based on (varying between one and eight single trials), iii) the CS+/CS– discrimination cutoff applied (varying between <0 μS and <0.1 μS), and iv) the CS+ type (only non-reinforced or all CS+ trials) considered. The different forking paths and their frequency resulting from these combinations are displayed in *Figure 2B*.

The cutoff for CS+ versus CS– discrimination used to identify a 'non-learner' varied between <0 μS and <0.1 μS, with most records excluding participants as 'non-learners' if they showed either a negative discrimination (<0 μS) and/or no discrimination (≤0 μS). These criteria apply if the SCR amplitude in response to the CS– was higher than and/or equal to the amplitude elicited by the CS +. Furthermore, most records required this criterion to be fulfilled only during the last half or the full fear acquisition training phase. Of note, the number of trials included in the same 'phase' category is contingent on the experimental design and hence does not represent a homogeneous category

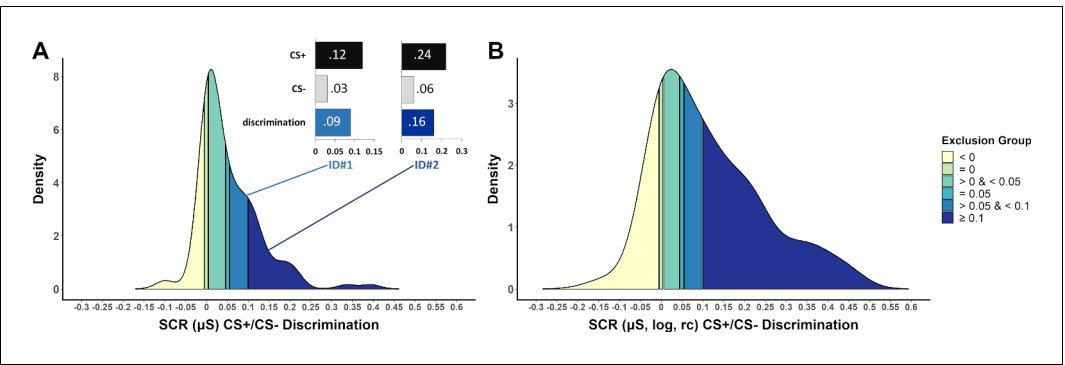

**Figure 3.** Density plots illustrating the frequency of CS+/CS– discrimination scores in a sample of *N* = 116 (Data set 1) based on the last half of the acquisition phase (including 7 CS+ and 7CS–, 100% reinforcement rate) for (**A**) SCR raw data and (**B**) logarithmized and range-corrected (rc; individual trial $SCR/SCR_{max\_across\_all\_trials}$) SCR data (as it is typically not reported to which data exclusion criteria are applied). Color coding (yellow to blue) illustrates which part of the sample would be excluded when applying the performance-based exclusion criteria (i.e. CS+/CS– discrimination) as identified by the systematic literature search. Panel (**A**) also illustrates two case examples (ID#1 and ID#2) that differ in SCR amplitudes but importantly show the same discrimination ratio between CS+ and CS– (4:1). These two case examples illustrate that high CS+/CS– discrimination cutoffs favor individuals with high SCR amplitudes to remain in the final sub-sample. Data are based on a re-analysis of an unpublished data set recorded in the fMRI environment (Klingelhöfer-Jens M., Kuhn, M. and Lonsdorf, T.B.; unpublished).

The online version of this article includes the following figure supplement(s) for figure 3:

**Figure supplement 1.** Percentages of participants excluded (Data set 1) when employing the different CS+/CS– discrimination cutoffs (as identified by the systematic literature search and graphically shown in *Figure 3B*) which are illustrated as density plots in *Figure 3*.

('last half' may include five trials for one study comprising 10 trials in total but 10 trials for a different study employing 20 trials in total.

## Applying the identified performance-based exclusion criteria to existing data: a case example

We applied the identified cutoff criteria to an existing data set (Data set 1) to exemplify the part of the sample that would be excluded when applying different cutoff criteria (shown in different colors from yellow to dark blue in *Figure 3*) based on the most frequently used phase restriction: the last half of fear acquisition training. CS+/CS– discrimination was calculated on the basis of raw (A) or log-transformed, range-corrected (log, rc) scores (B), because it is not usually reported which data are used to classify 'learners' vs. 'non-learners'. Strikingly, the proportion of participants that are excluded is higher when CS+/CS– discrimination is calculated on the basis of raw data rather than log-transformed and range-corrected data (despite employing the same criteria) in particular for the highest 'non-learner' <0.01 µS cutoff (76.7% versus 52.6%, respectively) (see *Figure 3—figure supplement 1* for details).

In addition, we included a case example of two hypothetical individuals that differ in raw SCR amplitudes (ID#1: low and ID#2: high), but importantly show the same discrimination ratio (4:1) between CS+ and CS–(see *Figure 3A*). These two case examples illustrate that high CS+/CS– discrimination cutoffs, such as excluding individuals with discrimination scores < 0.1 µS as 'non-learners', favor individuals with high SCR raw amplitudes.

Unsurprisingly, the exclusion group defined by a CS+/CS– discrimination cutoff <0 µS showed inverse discrimination (CS–>CS+, not significant in raw SCRs [p=0.117]; significant in log,rc SCRs [p = 0.021]). Strikingly and more importantly, most cumulative exclusion groups, as established by defining 'non-learners' by the CS+/CS– discrimination different cutoffs in SCRs in the literature, in fact show statistically significant CS+/CS– discrimination (see Appendix 2 for details and a brief discussion).

Note that despite the different color coding, which serves illustrative purposes only, the groups are in practice cumulative. More precisely, the groups illustrated by lighter colors are always

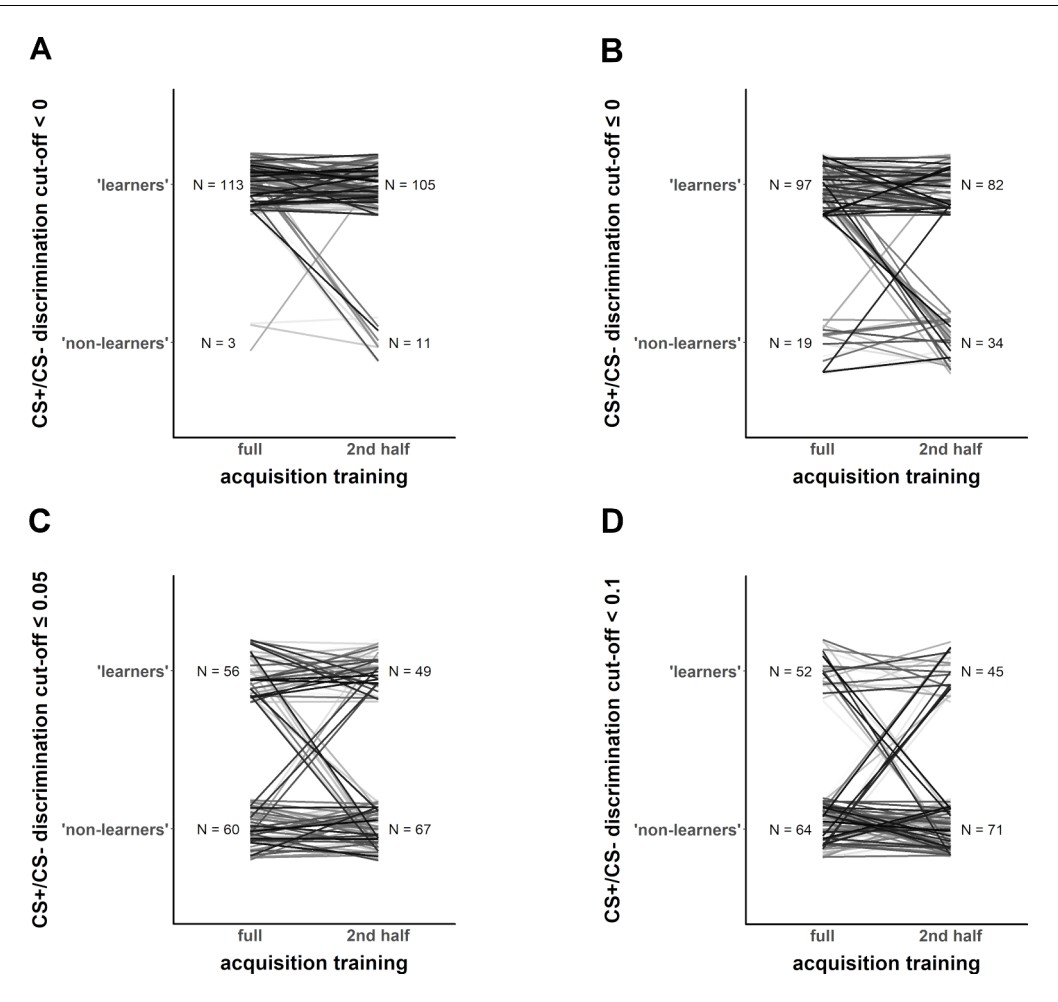

**Figure 4.** Exemplary illustration of individuals (Data set 1) that switch from being classified as 'learners' vs. 'non-learners' depending on the different CS+/CS– discrimination cutoff level (panels **A–D**), when calculation of CS+/CS– discrimination is based on either the full fear acquisition phase or the second half of the fear acquisition training (left and right part of each panel, respectively).

The online version of this article includes the following figure supplement(s) for figure 4:

**Figure supplement 1.** Bar plots (mean ± SE) on which the superimposed individual data points show CS+ and CS– amplitudes (of raw SCR values) and CS+/CS– discrimination in (**A**) fear ratings and (**B**) SCRs raw values in the group of 'non-learners', as exemplarily defined for this example as a group consisting of individuals in the two lowest SCR CS+/CS– discrimination cutoff groups (i.e., ≤0) in Data set 1.

contained in the darker colored groups when applying the respective cutoffs. For example, the group excluded when employing a cutoff of <0.1 µS (mid blue) also comprises the groups already excluded for the lower cutoffs of = 0.05 µS (light blue),<0.05 µS (turquoise), = 0 µS (light green) and <0 µS (yellow). For illustrative purposes, the different groups are treated as separate groups in this figure.

## Exploratory analyses of consistency of classification ('learners' vs. 'non-learners') across outcome measures and criteria employed

The convergence of non-discrimination across different outcome measures was investigated by testing for CS+/CS– discrimination in fear ratings in individuals with different amounts of CS+/CS– discrimination in SCRs as defined by the criteria described above. In fact, individuals with non-significant and inverse CS+/CS– discrimination (i.e., ≤0 µS) in SCRs showed significant CS+/CS– discrimination in fear ratings ($t_{31}$ = 9.69, $p_{bonf\_corr}$ < 0.000000001, $d$ = 1.71, see *Figure 4—figure*

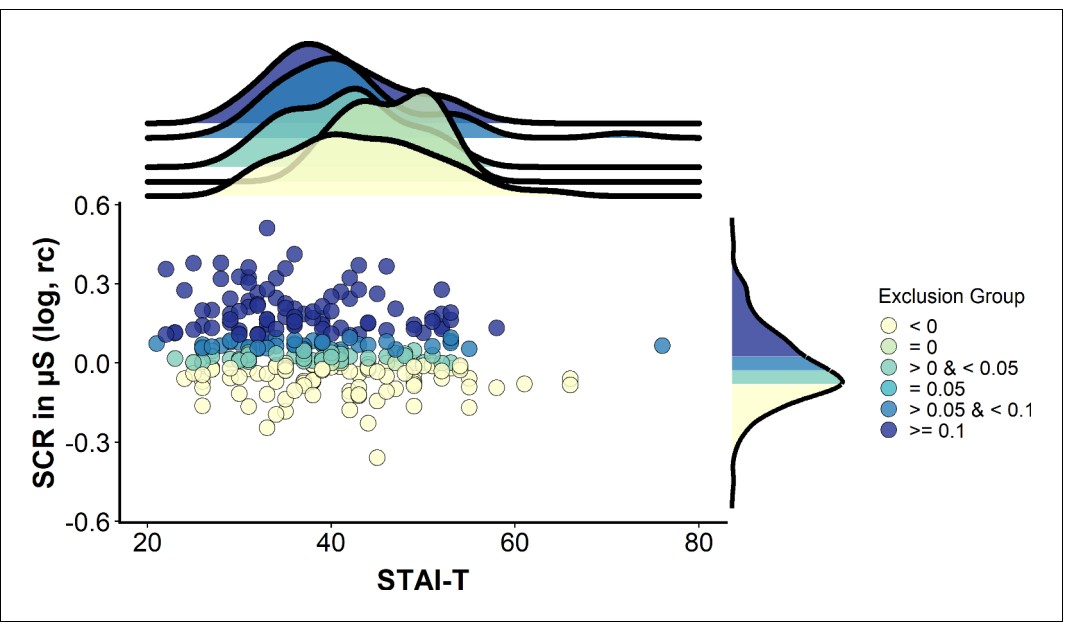

**Figure 5.** A case example illustrating potential sample bias induced by excluding individuals on the basis of CS+/CS– discrimination scores (based on logarithmized, range-corrected (rc) SCR data). Scatterplot illustrating the association between trait anxiety (measured via the trait version of the State-Trait Anxiety Inventory, STAI-T) and CS+/CS– discrimination scores in a sample of $N = 268$ (Data set 2). Color coding (yellow to blue) illustrates which part of the sample would be excluded when applying the performance-based exclusion criteria (i.e. CS+/CS– discrimination) as identified by the systematic literature search. Note that within this sample, no individuals were identified with CS+/CS– discrimination equaling 0.05 µS. The upper panel illustrates densities for trait anxiety for the different CS+/CS–discrimination groups. The rightmost panel illustrates the density for CS+/CS– discrimination in the full sample. Data are based on a re-analysis of a data set recorded in the behavioral environment (*Schiller et al., 2010*). Note that despite the different color coding, which serves illustrative purposes only, the groups are in practice cumulative. More precisely, the groups illustrated by lighter colors are always contained in the darker colored groups when applying the respective cutoffs. For example, the group excluded when employing a cutoff of <0.1 µS (mid blue) also comprises the groups already excluded for the lower cutoffs of = 0.05 µS (light blue), <0.05 µS (turquoise), = 0 µS (light green) and <0 µS (yellow). For illustrative purposes, the different groups are treated as separate groups in this figure.

*supplement 1*). Importantly, all cumulative exclusion groups showed significant CS+/CS– discrimination in fear ratings (all p 's< 0.002, see *Appendix 3—table 1*).

We also illustrate (*Figure 4*) that the classification as 'learners' and 'non-learners' changes if two features (CS+/CS- discrimination cutoff and full vs. last half of acquisition training phase) of the criteria are changed (as illustrated in their full variation in *Figure 2B*).

## The potential sample bias with respect to individual differences induced by employing different performance-based exclusion criteria: a re-analysis of existing data and a case example

Regarding the impact of performance-based exclusion on the pre-selection for certain individual differences, *Figure 5* shows that the distributions of trait anxiety were shifted to the left (i.e., towards lower scores) with higher SCR CS+/CS– discrimination cutoffs. More precisely, this means that, in this sample, highly anxious individuals display smaller CS+/CS– discrimination in SCRs, and that excluding individuals who display low discrimination scores will lead to the exclusion of anxious individuals.

In fact, we observed a main effect of 'Exclusion group' on trait anxiety score ($F_{[4,263]} = 219.2$, p<0.001, $\eta_P^2 = 0.77$). All exclusion groups (corresponding to the color coding in *Figure 5*) differ significantly from each other in their trait anxiety scores (all $p_{bonf\_corr} \leq 0.001$), except for the group that did not show any CS+/CS– discrimination (=0 µS, light green, however $n = 6$ only), which showed significantly higher trait anxiety scores (mean ± SD STAI score: 43.8 ± 6.1) than the group

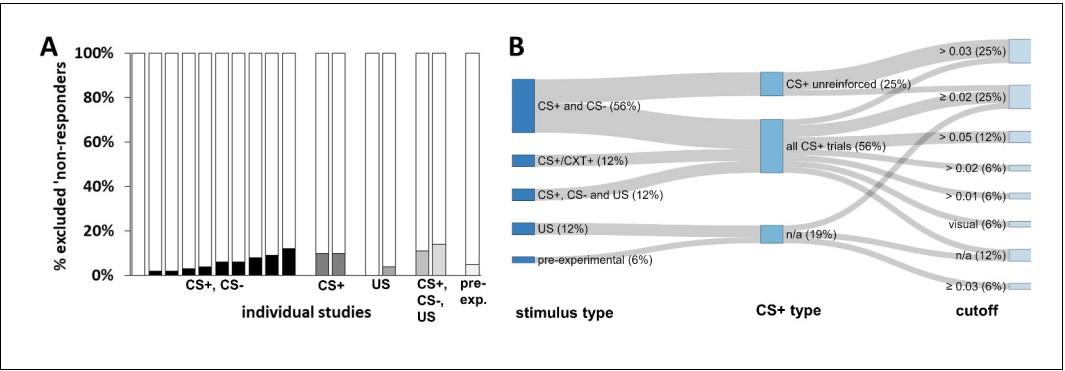

**Figure 6.** Graphical illustration of the percentage of 'non-responders' and forking path analysis across studies. (**A**) Illustration of the percentage of participants excluded from each study as a result of ' SCR non-responding' to (i) the conditioned stimuli (i.e., CS+ and CS–), (ii) the US, (iii) the CS+ (which also comprises a study that used the CXT+, i.e. context), (iv) the CS+, CS– and US or (v) a pre-experimental test. Note that these 18 individual studies are derived from 16 different records, two of which included two different studies that used the same criteria. Note that some studies excluded participants on the basis of 'non-learning' as well as 'non-responding', and hence the percentages displayed here do not necessarily map onto the percentage of total participants excluded from each study. Also note that a single study (*Schiller et al., 2018*) is not included in this visualization because it reported % 'non-learners' and % 'non-responders' as a single value. This value has been included in the visualization of 'non-learners' (*Figure 2*) as these are expected to represent the largest proportion. (**B**) Sanky plot illustrating the stimulus type (pre-experiment refers to determination of 'responding' in an unrelated phase prior to the experiment), the minimally required response amplitude in µS (note that 'visual' refers to visual inspection of the data without a clear-cut amplitude cutoff, NA refers to no criterion applied) illustrated as the nodes in graded colors and their combinations that lead to classification as a 'non-responder'. Path width was scaled in relation to frequency of the combinations. Note that for some 'nodes' the percentages do not add up to 100% because of rounding.

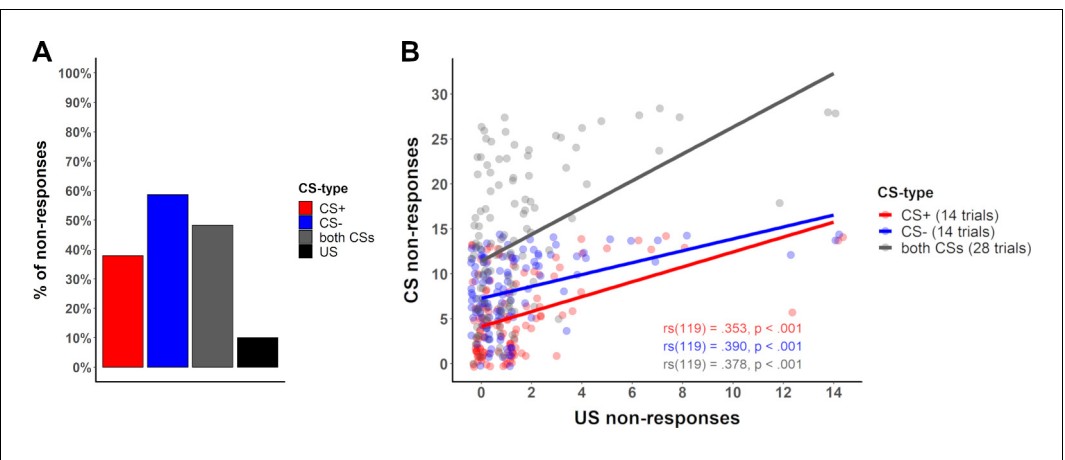

**Figure 7.** Percentage of no-responses across stimuli and correlation between CS and US non-responses. (**A**) Bar plot displaying the number of 'non-responses' to the CS+, CS–, across both CS and to the US across all participants in Data set 1 (see *Appendix 4—table 1* for percentages across different data sets). (**B**) Scatterplot illustrating the number of 'non-responses' (i.e., zero-responses, here defined by an amplitude <0.01 µS) to the US presentations (total of 14 presentations) and the CS+ (red) and CS– (blue) responses (14 presentations each) for each participant in Data set 1. For completeness sake, 'non-responses' across CS types are illustrated in gray (CS+ and CS– combined, total of 28 presentations). Lines illustrate the Spearman correlation (rs) between 'non-responses' to the US and 'non-responses' to the CS+, CS– and both CS, with corresponding correlation coefficients (font color corresponds to CS type) included in the figure.

with the largest CS+/CS– discrimination only (i.e., $\geq$0.1 µS, dark blue, $n$ = 88, mean ± SD STAI score: 36.6 ± 8.5, $p_{bonf\_corr} \leq$ 0.001, CI [0.133 to 0.279]). Nevertheless, trait anxiety scores in this group (light green) were not significantly larger than those in the group with the negative discrimination (i. e., <0 µS, yellow, $n$ = 89, mean ± SD STAI score: 40.5 ± 9.7, $p_{bonf\_corr}$ = 0.10, CI [−0.004 to 0.142]), the group with a small discrimination score (i.e.,>0 µS but <0.05 µS, light blue, $n$ = 43, mean ± SD STAI score 37.9 ± 7.9, $p_{bonf\_corr}$ = 1.0, CI [−0.054 to 0.094]) or the group with the middle discrimination score (i.e., >0.05 µS but <0.1 µS, mid blue, $n$ = 42, mean ± SD STAI score 38 ± 10.2, $p_{bonf\_corr}$ = 0.11, CI [−0.005 to 0.145]).

## Definition of 'non-responders' and numberof participants excluded across studies

Thirty-two percent (i.e., 16 records) of the records in our systematic literature search included a definition and exclusion of 'non-responders', with percentages of participants excluded as a result of non-responding ranging between 0% and 14% (see *Figure 6A*). A single study (*Chauret et al., 2014*; *Oyarzún et al., 2012*) reported % 'non-learners' and % 'non-responders' as a single value (see *Appendix 1—table 2*). The definitions differed in: i) the stimulus type(s) used to define 'non-responding' (CS+ reinforced, CS+ unreinforced, all CS+s, CS–, US), ii) the SCR minimum amplitude criterion used to define a responder (varying between 0.01 µS and 0.05 µS; visual inspection), and iii) the percentage of trials for which these criteria have to be met (see *Figure 6B* and *Appendix 1— table 2*), as well as a combination thereof.

'Non-responding' was most commonly defined as not showing a sufficient number of responses to the conditioned stimuli (CS+ and CS–), less frequently by the absence of responses to the US or any stimulus (CS+, CS- or US), and in two cases by the absence of responses to the CS+ or context (CXT+) specifically (see *Figure 6B*). Not surprisingly, the percentage of excluded participants differed substantially depending on the stimulus type used to define 'non-responding' (CS based, 0– 10%; CS+/CXT+ based, 10–11%; US based, 0–4%; CS and US based, 11–14%; pre-experimental test based, 5%; *Figure 6A*).

Despite these differences in the stimulus types used to define 'non-responding' in the first place, studies differed widely in the amplitude cutoff criterion to be exceeded in order to qualify as a response (see *Figure 6B*) as well as in the percentage of trials in which this cutoff had to be met (see *Appendix 1—table 2*).

The question of what (physiological) 'non-responders' during fear acquisition training are and how to identify them might be elucidated by investigating the number of 'non-responses' across trial types (CS and US) across data sets, and whether 'non-responding' to the US predicts 'non-responding' to the CS or vice versa. As expected from *Figure 6A*, the number of 'non-responses' to the US was low (as was also the case in our data [10%, Data set 1]), while the number of 'non-responses' to the CS (48.29%) was substantially higher – in particular for the CS– (58.6%; CS+ 'non-responses': 37.9%, see *Figure 7A*). This pattern, exemplarily illustrated here in one data set is representative of a larger number of data sets (see Appendix 4, table 1 for details). Furthermore, in our data (Data set 1), all individuals that did not react to the US in more than two thirds of the US trials also showed no responses to the CS ($n$ = 3 of $N$ = 119). To summarize, this provides the first evidence that 'non-responding' to the US may predict 'non-responding' to the CS but not vice versa. Furthermore, our data also suggest a positive correlation between the number of 'non-responses' to the US and the number of 'non-responses' to the CS (see *Figure 7B* for statistics).

## Discussion

In this article, we showed that participant exclusion in fear conditioning research is common (i.e., 40% of records included) and characterized by substantial operationalizational heterogeneity of definitions for 'non-learners' and (physiological) 'non-responders'. Furthermore, we provide case-examples that illustrate: i) the futility of some definitions of 'non-learners' (i.e., when those classified as 'non-learners' in fact show significant discrimination on both ratings and SCRs as illustrated in Appendix 3 and Appendix 2, respectively) when applied to our data; and ii) the potential sample bias induced by excluding 'non-learners' with respect to individual differences. Furthermore, we provide an overview of SCR 'non-responses' to different stimulus types (CS+, CS– and US) across different data sets (see *Appendix 4—tables 1* and *2*) as a guide for developing evidence-based criteria

# Box 1. List of reporting details, potential difficulties and recommendations when excluding 'non-learners' (performance-based exclusion) and/or 'non-responders' with a focus on SCRs.

Please note that this Box can be annotated online.

**(A) General reporting details**

| What to report? | Why is this considered important? | What can go wrong or be ambiguous? | Recommendations on how to proceed |
|---|---|---|---|
| Details on data recording and response quantification pipeline | • because differences in data recording and quantification (i.e., response scoring) can make a substantial difference | | • report recording equipment and all settings used (e.g., filter) <br>• report software used for response quantification <br>• report precise details of response quantification |
| Minimal response criterion (µS) to define a valid SCR | • to define valid responses | • minimally detectable amplitude (e.g., 0.01, 0.02, 0.03, 0.05 µS, etc.) may be sample- and equipment-specific <br>• no clear recommendations (existing guidelines provide a range of 0.01 to 0.05 µS) because this is influenced by noise level and equipment | • test different minimal response criteria in the data set and define the cutoff empirically. In our experience (Data set 1), a cutoff was easily determined empirically by visually inspecting responses at different cutoffs (e.g., <0.01 µS, between 0.01 µS and 0.02 µS) and by evaluating their discrimination from noise |
| Whether the first CS+ and/or the first CS– trial is included or not, and information on trial sequence | • no learning can be evident in the first trial, as the first US may occur at the earliest at the end of the CS+ and hence after the scoring window for the CS+-induced SCR <br>• if the first trial is a CS–, no learning can have taken place as the US has not been presented yet <br>• inclusion of the first trial (or the first trials in partial reinforcement protocols) may thus artificially reduce CS+/CS– discrimination | • in fully randomized partial reinforcement protocols, US presentations may cluster in the first or last half of the acquisition training, which will impact on CS+/CS–discrimination in SCRs | • careful experimental design with respect to trial-sequences (in particular in partial reinforcement protocols) <br>• report whether the first trial for both CS+ and CS– is excluded because it may induce noise and bias CS+/CS– discrimination towards non-discrimination and as the first trial is sensitive to trial sequence effects |
| Precise number of trials considered (if applicable for each trial type including reinforced and non-reinforced CS+ trials in case of partial reinforcement) | • often difficult/ambiguous to infer this information from the 'Materials and methods' section of a report[a] <br>• number of trials that the 'last half' or 'full phase' refers to is contingent on experimental design and hence ambiguous and imprecise (see **Figure 2B**) | | • precision in reporting rather than relying on the reader making the right inferences <br>• specify clearly the number of trials per stimulus type that are comprised in the 'last half' or 'full phase' <br>• provide a justification (theoretical and/or empirical) for this decision[b] |
| Details of whether results were based on raw or transformed data | • typically, transformations are required to allow interpretation of the reported results and to meet the assumptions of commonly statistical models | | • report details of transformation (e.g., logarithmized [log/LN], range-corrected, square-root) including the number of trials considered (for each stimulus type) and the sequence of transformations applied and specific formula (e.g., for range-correction) <br>• provide justification for any applied transformation (e.g., violation of assumption of normal distribution of residuals) |

| Precise number of excluded participants and specific reasons | • often difficult/ambiguous to infer this information from the 'Materials and methods' section of a report[a] | • different researchers have different opinions on what 'exclusion' is (e.g., having individuals discontinue after a first experimental day based on performance should be considered and reported as exclusion) | • report a breakdown of specific reasons for exclusions with respective $n$'s |

**(B) Specific reporting details for exclusion of 'non-learners'**

| What to report? | Why is this considered important? | What can go wrong or be ambiguous? | Recommendations on how to proceed |
|---|---|---|---|
| CS+/CS– discrimination is calculated on the basis of raw SCR or transformed (e.g., logarithmized [log/LN], range-corrected, square-root) scores | • the same criteria lead to different proportions of excluded individuals when applying them to raw or transformed data (see *Figure 3A and B*) | | • exact details of transformations (optimally calculation formulas) need to be included for full transparency and reproducibility |
| Minimal differential (CS+ vs. CS) cutoff for 'non-learning' in µS | • different cutoffs lead to very different proportions of individuals excluded (see *Figure 3*) | | • exact details on cutoffs need to be included for full transparency and reproducibility |
| On what outcome measures is 'non-learning' determined? | • 'non-learners' do not necessarily converge across different outcome measures (Appendix 3, *Figure 4—figure supplement 1*) | • all outcome measures recorded need to be reported | • 'non-learning' should not be based on a single outcome measure or a clear justification needs to be provided as to why a single measure is considered meaningful |
| If 'non-learning' is determined by responding during fear acquisition training, which trial types and number of trials per trial type were considered? | • depending on the criteria employed, the same individual may be classified as 'learner' or 'non-learner' (see *Figure 4*) | | • classification as 'non-learner' should be based on differential scores (CS+ vs. CS–), and the number of trials included for this calculation should be clearly justified. Providing a generally valid recommendation regarding the number of trials to be included is difficult because it critically depends on experimental design choices |
| If 'non-learning 'criteria are used, do they differ from criteria that the researcher or the research group used in previous publications? If yes, why were the criteria changed? | • provide explicit justifications on why different criteria were used previously and presently | | • report differences between present and previous criteria used including references and justifications |
| Did 'non-learners' really fail to learn? | • important as a manipulation check but note that the absence of a statistically significant CS+/CS– discrimination effect in a group on average cannot be taken to imply that all individuals in this group do not show meaningful CS+/CS– discrimination | • individuals classified as 'non-learners' may in fact show significant CS+/CS– discrimination in SCRs (see Appendix 2) or in other outcome measures (see *Figure 3—figure supplement 1* and Appendix 4) and hence fail the manipulation check | • do the groups classified as 'non-learners' and 'learners' differ significantly in discrimination, and do 'non-learners' really not discriminate in SCRs and other outcome measures? Report the data on this group graphically and/or statistically in the supplementary material (do not report the full sample with and without exclusions only) |

| | | | |
|---|---|---|---|
| Are results contingent on the exclusion of 'non-learners'? | • important to allow for transparency and to evaluate the impact of the results | • it is not clearly defined when results differ meaningfully when excluding and including 'non-learners' | • provide results with and without exclusion of 'non-learners' <br> • additional analyses can be provided as supplementary material. When results are not contingent on the exclusion of 'non-learners', it is sufficient to mention this briefly in the results of the main manuscript (e.g., results are not contingent on the exclusion of 'non-learners') <br> • if the results of the main analyses and hence the main conclusions change when 'non-learners' are excluded, this needs to be included in the main manuscript , and the implications need to be adequately discussed. Please note that this does not necessarily invalidate findings but can refine them |
| Descriptive statistics for excluded 'non-learners' | • important to allow for transparency and evaluation of the potential sample biases introduced | | • report sex, age, anxiety levels, awareness |

**(C) Specific reporting details for exclusions of 'non-responders'**

| What to report? | Why is this considered important? | What can go wrong or be ambiguous? | Recommendations on how to proceed |
|---|---|---|---|
| Whether 'non-responses' are calculated on the basis of raw SCR or transformed (e.g., logarithmized [log/LN], range-corrected, square-root) scores | • the same criteria lead to different proportions of excluded individuals when applying to raw or transformed data (see *Figure 3A and B*) | | • exact details of transformations (optimally calculation formulas) need to be included for full transparency and reproducibility |
| Minimal cutoff for 'non-responses' in µS | • it is often difficult/ambiguous to infer this information from the 'Materials and methods' section of a report[a] <br> • higher cutoffs could unnecessarily reduce the sample size | | • exact details on cutoffs need to be included for full transparency and reproducibility |
| Was 'non-responding' determined in a pre-experimental phase such as forced-breathing or US calibration? | • determining 'non-responding' during a pre-experimental phase may help to detect malfunctioning of the equipment and allow this to be corrected prior to data acquisition <br> • classification of 'non-responders' independent of the experimental task and its specifications (e.g., number of US presentations) | • electrodes may detach between the pre-experimental phase and fear acquisition training | • report details of pre-experimental phase <br> • classification in SCR 'non-responders' should be based on a pre-experimental phase if no US presentations occur during the experiment, such as in case of threat of shock experiments, observational conditioning, extinction or return of fear tests |
| If 'non-responding' is determined by responding during fear acquisition training, what trial types are considered? | • frequency of 'non-responding' differs substantially between different stimuli (CS and US) but also between CS+ and CS– (see *Figure 7A*) | • 'non-responding' to the US may be due to technical failure (i.e., no US was administered) | • classification in SCR 'non-responders' should **not** be based on SCRS elicited by CS (CS+, CS– or both), but should be based on US responding <br> • a question on the estimated number of US presented during fear acquisition training (and all other phases) may serve as a manipulation check |
| Descriptive statistics for excluded 'non-responder' | • important to allow for transparency and evaluation of the potential sample biases introduced | | • report sex, age, anxiety levels, awareness |

[a] based on our experience with extracting this information from literature identified in the systematic literature search reported in this manuscript.

[b] 'others have done this previously' is not an acceptable justification in our point of view.

to define 'non-responders'. Together, we believe that this work contributes to: i) raising awareness of some of the problems associated with performance-based exclusion of participants ('non-learners') and of how this exclusion is implemented, ii) facilitating decision-making on which criteria to employ and not to employ, iii) enhancing transparency and clarity in future publications, and thereby iv) fostering reproducibility and robustness as well as clinical translation in the field of fear conditioning research and beyond.

## 'Non-learners': conclusions, caveats and considerations

Operationalizational heterogeneity is illustrated by every single record in our systematic literature search (covering a six months period) that employed definitions of 'non-learners' using a set of idiosyncratic criteria. The true number of definitions in the field applied over decades will be even substantially larger. In the records included here, 6–52% of participants were excluded (disregarding one study reporting percentages of 'non-learners' and 'non-responders' together with 74%; cf. *Figure 2A*), which substantially exceed the percentages recently put forward for 'non-learning' exclusions (*Marin et al., 2019*) that were suggested to lie between 4% (*Chauret et al., 2014*) and 19% (*Oyarzún et al., 2012*).

If several thousand analytical pipelines can be applied, the likelihood of false positives is high (*Munafò et al., 2017*) and the temptation of their opportunistic (ab)use must be considered a threat. Hence, a constructive discussion on where to go from here and how to not get lost in the garden of forking paths is important. This being said, we do acknowledge that certain research questions or the use of different recording equipment (robust lab equipment vs. novel mobile devices such as smartwatches) may potentially require distinct data-processing pipelines and the exclusion of certain observations (*Silberzahn et al., 2018*; *Simmons et al., 2011*), and hence it is not desirable to propose rigid and fixed rules for generic adoption. Procedural differences, in particular the inclusion of outcome measures that require certain triggers to elicit a response (such as startle responses or ratings) have also been shown to impact on the learning process itself (*Sjouwerman et al., 2016*). Rather, we call for a reconsideration of methods in the field and want to raise awareness to the pitfalls of adopting exclusion criteria from previously published work without critical evaluation of whether these apply meaningfully to one's own research. Furthermore, we want to promote the adoption of transparent reporting of data processing, recording and analyses and strive to suggest standards in the field to reduce heterogeneity based on idiosyncratic customs rather than methodological and theoretical considerations (see *Box 1*).

Yet, there are many other critical considerations worth discussing beyond the heterogeneous criteria used to define 'non-learners' and their impact on the outcome of statistical tests:

First, 'performance-based exclusion of participants' is often based on a single outcome measure (typically SCRs), despite multiple measures being recorded (for exceptions see *Ahmed and Lovibond, 2019*; *Belleau et al., 2018*; *Oyarzún et al., 2012*). Importantly, 'fear learning' cannot be reliably inferred by means of SCRs, because SCRs capture arousal-related processes and can only be used as a proxy to infer 'fear learning' as fear is closely linked to arousal (*Hamm and Weike, 2005*). Relatedly, the fact that physiological proxies of 'fear' do not map onto 'fear' itself has been discussed extensively (*LeDoux, 2012*; *LeDoux, 2014*).

Second, but related, individuals that fail to show CS+/CS– discrimination in SCRs may show substantial discrimination, as an indicator of successful learning, in other outcome measures such as ratings of fear, US expectancy or fear potentiated startle (*Hamm and Weike, 2005*; *Marin et al., 2019*), as illustrated here for fear ratings (see *Figure 4—figure supplement 1* and *Appendix 3—table 1*).

Third, a common justification for excluding 'non-learners' is that it is not possible to investigate extinction- or return-of-fear-related phenomena in individuals who 'did not learn'. To our knowledge, there is some evidence (*Craske et al., 2008*; *Plendl and Wotjak, 2010*; *Prenoveau et al., 2013*) that this theoretical assumption does not necessarily hold true, (i.e., CS+/CS– discrimination during fear acquisition training does not necessarily predict CS+/CS– discrimination during other experimental phases) (*Gerlicher et al., 2019*). An empirical investigation of this, however, would go beyond this manuscript's scope.

Fourth, we provided empirical evidence that those classified as a group of 'non-learners' in SCRs in the literature (sometimes referred to as 'outliers') on the basis of the identified definitions in fact displayed significant CS+/CS– discrimination when applied to our own data. An exception to this

was using cut offs in differential responding of <0.05 μS (note, however, that a non-significant CS+/CS– discrimination effect in the group of 'non-learners' as a whole cannot be taken as evidence that all individuals in this group do not in fact display meaningful or statistically significant CS+/CS– discrimination). Hence, in addition to the many conceptual problems we raised here, the operationalization of 'non-learning' in the field failed its critical manipulation check given that those classified as 'non-learners' show clear evidence of learning as a group (i.e., CS+/CS– discrimination, see *Appendix 2—table 1*).

Fifth, we illustrate a concerning sample bias that is introduced by performance-based participant exclusion. CS+/CS– discrimination in SCRs during fear acquisition training has been linked to a number of individual difference factors (*Lonsdorf and Merz, 2017*) and, naturally, selecting participants on the basis of SCR CS+/CS– discrimination will also select them on the basis of these individual differences (illustrated by our case example on trait anxiety, *Figure 5*). In our case example, we illustrate that excluding 'non-learners' biases the sample towards low anxiety scores, which hampers the generalizability and replicability of findings: i) the effect may only exist in low-anxiety individuals but not in the general population, and ii) as fear acquisition is a clinically relevant paradigm, pre-selection in favor of low-anxiety individuals might represent a threat to the clinical translation of the findings. Many studies in the field of fear conditioning research aim to develop behavioral or pharmacological manipulations to enhance treatment effects or aim to study mechanisms that are relevant for clinical fear and anxiety. Hence, it is highly problematic that these studies may exclude individuals who show response patterns that mimic responses typically observed in anxiety patients when excluding 'non-learners'. In fact, patients suffering from anxiety disorders have been shown to be characterized by generalization of fear from the CS+ to the CS– (*Duits et al., 2015*).

Sixth, as illustrated by our case example (*Figure 3*), high CS+/CS– discrimination cutoffs generally favor individuals with high SCR amplitudes despite potentially identical ratios between CS+ and CS– amplitudes, which may introduce a sampling bias for individuals characterized by high arousal levels that probably have biological underpinnings. Relatedly, future studies need to empirically address which criteria for SCR transformation and exclusions are more or less sensitive to baseline differences (for an example from startle responding see *Bradford et al., 2015*; *Grillon and Baas, 2002*).

In summary, in light of the many (potential) problems associated with performance-based exclusion of participants, we forcefully echo Marin et al.'s conclusion that one needs "to be cautious when excluding SCR non-learners and to consider the potential implications of such exclusion when interpreting the findings from studies of conditioned fear" (*Marin et al., 2019*, abstract). Routinely, excluding participants who are intentionally or unintentionally characterized by specific individual differences represents a major threat to generalizability, replicability and potentially clinical translation of findings, as results might be contingent on a specific sub-sample and specific sample characteristics. This is also true when researchers are interested in the study of general processes. Furthermore, by excluding these individuals from further analyses, we may miss the opportunity to understand why some individuals do not show discrimination between the CS+ and the CS– in SCRs (or other outcome measures) or whether this lack of discrimination is maintained across subsequent experimental phases. It can be speculated that this lack of discrimination may carry meaningful information – at least for a subsample.

## 'Non-responders': conclusions, caveats and considerations

In addition to 'non-learners', 'non-responders' are also often excluded during fear conditioning research. We showed that the definition of 'non-responders', like that of 'non-learners', varies widely across studies. Heterogeneity in definitions manifests in different cutoff criteria for what is considered a valid response, the number of trials and the stimulus type(s) considered (*Appendix 1—table 2*, *Figure 6*). Surprisingly, most definitions are based on CS responses (i.e., SCRs to the CS+ and/or CS–) and only few are based on US responses. This highlights a potentially problematic overlap between 'non-learners' and 'non-responders': 'non-responding' to the CS (i.e., CS+ and CS– or CS+ only) is not necessarily indicative of physiological 'non-responding' – especially if high cutoffs are used. In fact, 'non-responding' to the CS may, or at least in some cases, reflect the absence of learning-based patterns in physiological responding – which may carry important information. Having observed the striking differences in percentages of 'non-responses' to the US (10%) and CS (48%) observed in our data (see *Figure 7* and *Appendix 4—table 1*), we suggest that physiological

'non-responding' cannot and should not be determined on the basis of the absence of responding to the CS.

More globally, the group of 'non-responders', as defined by the criteria identified here, probably lumps together several sub-groups: individuals (1) for whom technical problems resulted in no valid SCRs, (2) who fell asleep or did not pay attention, (3) who cognitively learned the CS+/US contingencies but did not express the expected corresponding responses in SCRs, and (4) who were attentive to the experiment but did not learn the contingencies (i.e., unaware participants) and hence did not show the expected SCR patterns (*Tabbert et al., 2011*).

In summary, although excluding physiological 'non-responders' makes sense (in terms of a manipulation check and independent of the hypothesis), we consider defining 'non-responders' on the basis of the absence of SCRs to the CS as problematic (dependent on the hypothesis). We suggest that physiological SCR 'non-responders' should be defined on the basis of US responses during fear acquisition training or to strong stimuli during pre-conditioning phases such as US calibration, startle habituation or forced breathing (reliably eliciting strong SCRs). If 'non-responding' to the US (during fear acquisition training) is used, it is difficult to suggest a universally valid cutoff with respect to the number or percentage of required valid US responses, because this critically depends on a number of variables such as hardware and sampling rate used. It remains an open question for future work whether data quality of novel mobile devices (e.g., smartwatches) for the acquisition of SCRs differs from traditional, robust lab-based recordings and how this would impact on the frequency of exclusions based on SCRs. Appendix 4 suggests that the cutoff may typically range between 1/3 and 2/3 of valid responses but may be data-set specific. US-based criteria are of course not trivial in multiple-day experiments, in which certain experimental days do not involve the presentation of US or involve few temporally clustered US presentations (i.e., reinstatement), or in paradigms not involving direct exposure to the US (i.e., observational or instructional learning; *Haaker et al., 2017*). In these cases, the other options listed above are strongly preferred to CS based criteria.

## Where do we go from here?

In this work, we have comprehensively illustrated and argued that most of the current definitions employed to define 'non-learners' and 'non-responders' have to be considered as theoretically and empirically problematic. It is not sufficient, however, to raise awareness to these problems and the practical question of 'Where do we go from here?' remains to be addressed. What can we do to avoid getting lost in the garden of forking paths of exclusion criteria? Here, we would like to offer several solutions to improve practices in the field, which we expect to foster robustness, replicability and potentially clinical translation of findings: (1) transparency in reporting, (2) adopting open science practices, (3) increasing the level and quality of reporting and (4) graphical data presentation, (5) manipulation checks, and (6) fostering critical evaluation. We refer to see *Box 1* for specific recommendations.

More precisely, **transparency** can be enhanced 'if observations are eliminated, authors must also report what the statistical results are if those observations are included', as suggested by Simmons and colleagues, nearly a decade ago (*Simmons et al., 2011*, Table 2). Here, we echo this call that this recommendation should be implemented routinely in data reporting pipelines when employing performance-based participant exclusions ('non-learners') in fear conditioning research. We also call for a transparent and adequate reporting in the results (in brief) and discussion section rather than providing this information exclusively in the appendix. This being said, it is important to point out that should a finding turn out to be contingent on the exclusion of 'non-learners', this does not necessarily invalidate this finding. On the contrary, it may further specify the finding or hint to possible mechanisms and/or boundary conditions – yet inferences on boundary conditions should be made carefully (*Hardwicke and Shanks, 2016*). Relatedly, adopting an **open science culture** will facilitate transparent reporting of exclusion criteria (*Nosek et al., 2015*) and will minimize the risk of exploiting heterogeneous definitions in the field. Registered reports (*Hardwicke and Ioannidis, 2018*), publicly available data including those from excluded participants and pre-registration (*Munafò et al., 2017*) of definitions and analysis pipelines (*Ioannidis, 2014*), as well as openly accessible lab-specific standard operational protocols (SOPs), may also be helpful.

We acknowledge, however, that transparent reporting and particularly pre-registration of exclusion criteria is not trivial in light of the unsatisfactory **quality and level of detail in reporting** in the field of fear conditioning research. It was striking that the compilation of exclusion criteria ('non-

learners' and 'non-responders', see *Appendix 1—tables 1* and *2*) employed in the records included in our systematic literature search required extensive personal exchange with the authors because the definitions provided were often insufficient, ambiguous or incorrect. It is our responsibility as authors, reviewers and editors to improve these reporting standards to an acceptable level. As a guidance, *Box 1* provides a compilation of reporting details that we consider important to include in both pre-registered protocols and publications (an editable online version of *Box 1* is available to allow for further development, see Box caption).

Our recommendations to improve the level of reporting details and transparency extends to the **graphical illustration of results**, which should optimally allow for a complete presentation of data (*Weissgerber et al., 2015*) without risking obscuring important patterns, providing detailed distributional information rather than merely presenting summary statistics (see *Weissgerber et al., 2015* for a discussion). Such visualization options include, for instance, scatterplots, box plots, histograms, violin plots as well as their combination (see also *Figure 5*) in so called 'rain cloud plots' (see *Allen et al., 2018* for a tutorial in R, Matlab and Phyton) and utilizing colors or color gradients to visualize different groups of individuals (for instance 'learners' and 'non-learners') or discrimination scores. This will provide readers with the opportunity to evaluate the presented results and conclusions independently and comprehensively.

Finally, if criteria for 'non-learners' or 'non-responders' are employed to exclude participants from data analyses (or continuation of the experiment), we recommend that a **sanity or manipulation check** should be performed to determine whether – for instance - 'non-learners' really did not learn (i.e., really do not show significant CS+/CS– discrimination). We have empirically illustrated that most definitions of 'non-learners' fail this manipulation check (*Appendix 2—table 1*). Yet, it may not be feasible in all cases to determine such statistics, as these may not be appropriate for small samples and correspondingly small sub-groups of 'non-learners'. Relatedly, we urge authors to justify adequately all details of the exclusion criteria (if applied) – both theoretically and practically. Furthermore, we encourage authors, reviewers and editors alike to **critically evaluate** whether exclusions and applied criteria are warranted in the first place and appropriate in the specific context (vs. mere adopting published or previously employed criteria) and whether these exclusion criteria are transparently reported and discussed if results hinge on them (*Steegen et al., 2016*).

Furthermore, future work should empirically address the question of how to best define 'non-learning' in particular in light of different outcome measures in fear conditioning studies, which capture different aspects of defensive responding (*Jentsch et al., 2020*; *Lonsdorf et al., 2017*).

## Final remarks

In closing, the field of fear conditioning has been plagued with a lack of consensus on how to define and treat 'non-learners' and 'non-responders', which not seldomly impacts review processes and generates unnecessary lengthy discussions for editors, reviewers and authors. We argue that it is neither ethical (due to an excessive waste of tax money and human resources) nor scientifically meaningful to exclude up to two thirds of a sample. If only one third of the population performs 'as expected' in the experiment, experimental designs, data recording and processing techniques as well as definitions need to be reconsidered. We have shown that findings derived from such highly selective sub-samples may not generalize to other samples or to the general population, and as a consequence might be a threat to clinical translation. Most problematically, however, findings derived from such highly selective samples have been routinely and invariantly generalized to reflect 'general principles' and 'processes' in the past. Not surprisingly, such findings have also suffered replication failures. As such, exclusions of 'non-learners' can in fact be dangerous if not handled transparently (as suggested above), because they may bias and confuse a whole research field and may push research along a misleading path. Thus, we suggest recommendations and consensus suggestions, and recommend that common practices should be critically evaluated before we adopt them in future work, so that the field follows a path towards more robust and replicable research findings.

## Materials and methods

This project has been pre-registered on the Open Science Framework (OSF) (*Lonsdorf et al., 2019*, March 22; retrieved from https://osf.io/vjse4).

A **systematic literature search** was performed according to PRISMA guidelines (*Moher et al., 2009*) covering all publications (including e-pubs ahead of print) in PubMed during the six months prior to the 22$^{nd}$ March 2019, using the following search terms: threat conditioning OR fear conditioning OR threat acquisition OR fear acquisition OR threat learning OR fear learning OR threat memory OR fear memory OR return of fear OR threat extinction OR fear extinction. In case of author corrections, we included the original study that the correction referred to unless this study itself was already included on the basis of the publication date.

From the identified 854 records listed in PubMed, 152 were included in stage 2 screening (abstract) and 86 were retained for stage 3 screening (full text). Finally, 50 records were included (see *Figure 1* for details) that reported results for (1) SCRs as an outcome measure from (2) the fear acquisition training phase (3) in human participants.

### Extraction of criteria for 'non-learners' and 'non-responders'

The 50 records were screened in-depth and information derived from each record was entered into a template file agreed on by the authors prior to literature screening (available from the OSF pre-registration https://osf.io/vjse4). We distinguished between 'non-learners' and 'non-responders'. We considered an exclusion to be an exclusion of 'non-learners' if it was based on the key task performance –that is, CS+/CS– discrimination in SCRs. Exclusions were considered as exclusion of 'non-responders' when based on general (physiological) responding (i.e., *not* based on CS+/CS– discrimination). Participants who were explicitly excluded because of clear-cut and well-described technical problems, such as abortion of data recording or electrode disattachment, were not included in any definition. Criteria for defining 'non-learners' (see *Appendix 1—table 1*) and 'non-responders' (see *Appendix 1—table 2*) were extracted if applicable for the respective study. In case information in the publication was insufficient or ambiguous, the corresponding authors were contacted and asked for clarification.

### Re-analysis of existing data applying the identified exclusion criteria

One aim of this work was to illustrate empirically the impact of different exclusion criteria on the study outcome and interpretation. To achieve this aim, we initially planned to re-analyze existing data sets and to exclude participants on the basis of the identified definitions, which was expected to demonstrate that results are not robust across the various definitions of 'non-learners' and 'non-responders' employed. More precisely, we planned to calculate CS+/CS– discrimination across different data sets for all definitions identified by the systematic literature search and to generate corresponding correlation matrices as well as the percentages of zero and non-responses (see pre-registration: https://osf.io/vjse4). Because the exclusion criteria identified through the systematic literature search were even more heterogeneous than expected, and as it was difficult to agree on a key outcome to quantify the impact of exclusion criteria, we eventually concluded that such extensive re-analyses would not add much to the tabular and graphical illustration of this heterogeneity. Instead, we provide illustrative case examples for: (i) the proportion of individuals excluded on the basis of the identified exclusion criteria for 'non-learners' (*Figure 3*) and (ii) the potential sample bias with respect to individual differences (exploratory aim) induced by employing different exclusion criteria features (i.e., discrimination cutoff; *Figure 5*). As planned, (iii) we provide the percentage of non-responses to the CS+, CS–, CS+ and CS– combined, and the US across different studies, as well as empirical information on the association between CS and US based non-responding as a base to guide empirical recommendations.

Data processing, statistical analyses and figures were generated with R version 3.6.0 (2019-04-26) using the following packages: cowplot, dplyr, ggplot2 (*Wickham, 2009*), ggrigdes, car, ez, lsr, psychReport, lubridate, RColorBrewer and flipPlot packages. Sanky plots were generated with help of https://app.displayr.com.

## Data sets

### Data set 1

Data set 1 is part of the baseline measurement of an ongoing longitudinal fear conditioning study. Here, fear ratings and SCR data from the first test-timepoint ($T_0$) were included (N = 119, 79 females, mean ± SD age of 25 ± 4 years) whereas fMRI data were not used. All participants gave written informed consent to the protocol which was approved by the local ethics committee (PV 5157, Ethics Committee of the General Medical Council Hamburg).

Data set 1 is employed to illustrate a case example for the proportion of participants excluded when employing different CS+/CS– discrimination cutoffs ('non-learners', *Figure 3*) as well as the number of zero-responses across different stimulus types ('non-responders') and their association (*Figure 7*). Furthermore, we aimed to test exploratively whether even in groups defined as 'non-learners' a significant CS+/CS– discrimination on SCR and fear ratings can be detected (all results presented in the Appendix are based on Data set 1).

### Paradigm and stimuli

The two-day paradigm consisted of habituation and acquisition training (day 1) and extinction training and recall testing (day 2) without any contingency instructions provided. Here, only data from the acquisition training phase (100% reinforcement rate) were used. CS were two light grey fractals, presented 14 times each in a pseudo-randomized order for 6–8 s (mean: 7 s). Visual stimuli were identical for all participants, but allocation to CS+ and CS– was counterbalanced between participants. During inter-trial intervals (ITIs), a white fixation cross was shown for 10–16 s (mean: 13 s). All stimuli were presented on a light gray background and controlled by Presentation software (Version 14.8, Neurobehavioral Systems, Inc, Albany California, USA).

The electrotactile stimulus, serving as US, consisted of three 10 ms electrotactile rectangular pulses with an interpulse interval of 50 ms (onset: 200 ms before CS+ offset) and was administered to the back of the right hand of the participants. It was generated by a Digitimer DS7A constant current stimulator (Welwyn Garden City, Hertfordshire, UK) and delivered through a 1 cm diameter platinum pin surface electrode (Speciality Developments, Bexley, UK). The electrode was attached between the metacarpal bones of the index and middle finger. US intensity was individually calibrated in a standardized step-wise procedure aiming at an unpleasant, but still tolerable level.

### SCRs

SCRs were semi-manually scored by using a custom-made computer program (EDA View) as the first response from trough to peak 0.9–3.5 s after CS onset (0.9–2.5 s after US onset) as recommended (*Boucsein et al., 2012*; *Sjouwerman and Lonsdorf, 2019*). The maximum rise time was set to 5 s. Data were down-sampled to 10 Hz. Each scored SCR was checked visually, and the scoring suggested by EDA View was corrected if necessary (e.g., the foot or trough was misclassified by the algorithm). Data with recording artifacts or excessive baseline activity (i.e., more than half of the response amplitudes) were treated as missing data points and excluded from the analyses. SCRs below 0.01 µS or the absence of any SCR within the defined time window were classified as non-responses and set to 0. The threshold of 0.01 µS for this data set was determined empirically by visually inspecting response specifically above and below this cutoff, which suggested that in this data set, responses > 0.01 µS can be reliably identified. 'Non-responders' (N = 3) were defined as individuals who showed more than two thirds of non-responses to the US (10 or more non-responses out of 14 US trials, see *Appendix 4—table 2*). Three individuals were classified as 'non-responders' and these individuals did not show any responses to the CS either. The three participants classified as 'non-responders' (see above) were only excluded for the analyses of 'non-learners'. Raw SCR amplitudes were normalized by taking the natural logarithm and range-corrected by dividing each logarithmized SCR by the maximum amplitude (maximum SCR to a CS or a US) per participant and day.

### Fear ratings

Fear ratings were provided by participants through ratings on a visual analog scale (VAS) on the screen asking 'how much stress, fear, and tension' they experienced when they last saw the CS+ and CS–. The fear ratings used for the purpose of this manuscript are those obtained after fear acquisition training (no ratings were acquired during this phase). Answers were given within 5 s on the VAS,

which ranged from 0 (answer = none) to 25 (answer = maximum) by using a button box. Pressing the buttons moved a bar on the VAS to the aimed value and answers were logged in by pressing another button. Non-registered ratings were considered as missing values (8.4%).

## Statistical analysis

To test whether exclusion groups differ in CS+/CS– discrimination, a mixed ANOVA with CS+/CS– discrimination in SCR or fear ratings as the dependent variable and the between-subjects factor 'Exclusion group' and the within-subject factor 'CS-type' was performed. Note that it is circular to test for differences in SCR CS+/CS– discrimination between groups that were selected on the basis of different SCR CS+/CS– discrimination cutoffs in the first place. Still, it is relevant to test whether all groups classified as 'non-learners' in the literature do indeed fail to show evidence of learning, which would be indicated by a lack of significant CS+/CS– discrimination in SCRs in this case. In essence, this is a test to evaluate whether the exclusion criteria used in the literature do indeed achieve what they purport to do, that is, classify a group of participants that do not show evidence of learning. To test whether these exclusion groups discriminated in SCRs and fear ratings, exclusion groups were cumulated, and $t$-tests were performed for each cumulative group (see Appendices 2 and 3, respectively). We acknowledge, however, that the absence of a statistically significant CS+/CS– discrimination effect in a group on average cannot be taken to imply that all individuals in this group do not show meaningful CS+/CS– discrimination. As such, this is a rather conservative test. To correct for multiple testing, all $p$-values deriving from $t$-tests were adjusted using the Bonferroni procedure. As effect size, Cohen's $d$ was reported for $t$-tests and partial eta-squared for ANOVAs. To illustrate the association between the non-responses to the US and the non-responses to the CS, a Spearman rank correlation test was computed.

## Data set 2

For the purpose of this manuscript, a final sample of 268 individuals (195 female, mean ± SD age of 25 ± 4 years) was re-analyzed. This sample is reported in a recent pre-print (*Sjouwerman et al., 2018*) in which we observed an association between trait anxiety and CS+/CS– discrimination in SCRs. Here, the re-analysis and graphical illustration of these data serve the purpose of a case example to illustrate the potential sample bias that may be induced by employing performance-based exclusion (*Figure 5*).

## Paradigm and stimuli

A detailed experimental description is included in the preprint *Sjouwerman et al. (2018)*. In brief, participants underwent a 100% reinforcement fear acquisition training phase in a behavioral laboratory setting, including 9 CS+ and 9 CS– trials, presented for 6 s each. Consequently, 9 US presentations were included that coincided 100 ms prior to CS+ offset. Trials were interleaved by 10–13 s ITIs with a white fixation cross presented on a black background. Black geometrical shapes served as CS, and electrical stimulation delivered by a DS7A electrical stimulator (Digitimer, Welwyn Garden City, UK) onto the outer surface of the right hand served as US. The intensity of the US was individually calibrated with a stair-case procedure in order to reach an unpleasant but tolerable level. Not of interest to the current case example were the acoustic startle probes (95 dB(A) burst of white noise) presented to elicit a startle response in two thirds of all acquisition trials, as well as three fear-rating blocks probed intermittently during fear acquisition training. Startle probes were presented 4 or 5 s post-CS onset, and 5 or 7 s post-ITI onset. No contingency instructions were given.

## SCRs

SCRs were quantified as the first SCR within 0.9–4.0 s after stimulus onset (CS or US) and were scored semi-manually from trough-to-peak using a custom-made program. Signal increases smaller than 0.02 μS were treated as non-responses, that is set to 0. (Please note that this cut-off was not empirically determined as in Data set 1 but adopted from the previous publication of Data set 2. As we present re-analyses here, we decided not to change the cut-off to maintain comparability.) Responses confounded by recording artifacts, such as responses moving beyond the sampling window, excessive baseline activity, or electrode detachment were treated as missing values. Raw response amplitudes per trial were log-transformed and range-corrected for the maximum CS or US response per participant. Individuals not showing any valid SCR (i.e., missing or zero responses) in

more or equal than two thirds (≥6 out of 9, see *Appendix 4—table 2*) of US trials were treated as physiological 'non-responders' (*n* = 19) and were consequently excluded from graphical illustration and the statistical analysis. In addition, 31 participants were excluded prior to physiological processing, either because of abortion of the experiment or due to technical failures during data acquisition (e.g. errors during saving, overwritten logfile, or missing markers), leaving 307 out of 357 individuals with valid SCR data for fear acquisition training. Of these 307 participants, 39 had incomplete STAI-T data (*Spielberger et al., 1983*) resulting in a final sample size for this case example of 268 individuals (195 female, mean ± SD age of 25 ± 4 years).

## Statistical analysis

To test whether different exclusion groups differ in their mean trait anxiety levels, a univariate ANOVA with STAI-T score as the dependent variable and exclusion group as the independent variable was carried out. Post hoc pairwise *t*-tests were conducted to compare trait anxiety scores between the different exclusion group levels. The post hoc tests were corrected for multiple testing, using the Bonferroni correction method. 95% family wise confidence levels were determined using TukeyHSD tests.

# Acknowledgements

The project is part of a European wide network (EIFEL-ROF network) of researchers working on meta-research topics in the field of fear conditioning research, which is funded by the German Research foundation (DFG; Grant ID LO1980/2-1) to TBL and CJM. Data used for re-analyses in the main manuscript paper are derived from projects funded by the DFG to TBL (Data set 1: Collaborative Research Center project number 44541416 TRR 58, sub-project B07. Data set 2: LO 1980/1-1). Data reported in *Appendix 1—table 1* are also funded by the German Research Foundation to CJM (Collaborative Research Center SFB 1280 project number 316803389, sub-project A09) and JW (WE 5873/1–1 and WE 2762/5–1). The authors represent eight different research groups across three European countries (Germany, the Netherlands, and Belgium).

We thank Prof. Dr. Matthias Gamer, University of Würzburg, for providing EDA View for SCR response quantification.

The authors thank Manuel Kuhn, Jan Haaker, Tanja Jovanovic and Dean Mobbs for helpful suggestions during discussions on this project. We also thank Jan Haaker for help with initial literature screening.

# Additional information

## Funding

| Funder | Grant reference number | Author |
| --- | --- | --- |
| Deutsche Forschungsgemeinschaft | LO 1980/2-1 | Tina B Lonsdorf<br>Christian J Merz |
| Deutsche Forschungsgemeinschaft | LO 1980/1-1 | Tina B Lonsdorf |
| Deutsche Forschungsgemeinschaft | Project B07 44541416 | Tina B Lonsdorf |
| Deutsche Forschungsgemeinschaft | 316803389 - SFB1280 | Christian J Merz |
| Deutsche Forschungsgemeinschaft | WE 5873/1-1 | Julia Wendt |
| Deutsche Forschungsgemeinschaft | WE 5873/5-1 | Julia Wendt |

The funders had no role in study design, data collection and interpretation, or the decision to submit the work for publication.

## Author contributions

Tina B Lonsdorf, Conceptualization, Data curation, Formal analysis, Supervision, Funding acquisition, Visualization, Methodology, Writing - original draft, Project administration, Writing - review and editing, Concieved the study, Organized the project group, Contributed to data acquisition, Data analysis (systematic literature search, Dataset 1, Dataset 2, Data visualization, Drafting the manuscript, Revised the draft critically and approved the final version to be published; Maren Klingelhöfer-Jens, Data curation, Formal analysis, Visualization, Writing - original draft, Writing - review and editing, Contributed to data analysis and data visualization (data set 1), Drafting the manuscript, Revised the draft critically and approved the final version to be published; Marta Andreatta, Tom Beckers, Anastasia Chalkia, Anna Gerlicher, Valerie L Jentsch, Shira Meir Drexler, Gaetan Mertens, Jan Richter, Julia Wendt, Formal analysis, Writing - original draft, Writing - review and editing, Contributed to data acquisition and data analysis (systematic literature search), Drafting the manuscript, Revised the draft critically and approved the final version to be published; Rachel Sjouwerman, Data curation, Formal analysis, Visualization, Writing - original draft, Writing - review and editing, Contributed to data acquisition, Data analysis (dataset 2), Data visualization, Drafting the manuscript, Revised the draft critically and approved the final version to be published; Christian J Merz, Conceptualization, Formal analysis, Supervision, Funding acquisition, Writing - original draft, Writing - review and editing, Conceived the study, Contributed to data acquisition and data analysis (systematic literature search), Drafting the manuscript, Revised the draft critically and approved the final version to be published

## Author ORCIDs

Tina B Lonsdorf (iD) https://orcid.org/0000-0003-1501-4846
Marta Andreatta (iD) https://orcid.org/0000-0002-1217-8266
Tom Beckers (iD) https://orcid.org/0000-0002-9581-1505
Anastasia Chalkia (iD) https://orcid.org/0000-0002-1613-2281
Valerie L Jentsch (iD) https://orcid.org/0000-0001-9318-9540
Shira Meir Drexler (iD) https://orcid.org/0000-0001-8797-6900
Jan Richter (iD) https://orcid.org/0000-0002-7127-6990
Julia Wendt (iD) https://orcid.org/0000-0003-2299-5881
Christian J Merz (iD) https://orcid.org/0000-0001-5679-6595

## Ethics

Human subjects: Study 1: All participants gave written informed consent to the protocol which was approved by the local ethics committee (PV 5157, Ethics Committee of the General Medical Council Hamburg). Study 2: All participants gave written informed consent to the protocol which was approved by the Ethical Review Board of the German Psychological Association (TL072015).

## Decision letter and Author response

Decision letter https://doi.org/10.7554/eLife.52465.sa1
Author response https://doi.org/10.7554/eLife.52465.sa2

# Additional files

## Supplementary files

• Transparent reporting form

## Data availability

The minimal data sets (data set 1 and data set 2, both represent re-analysis of existing data), which were analysed during the current study, as well as code for figure production are are available at OSF under https://osf.io/mkxqe/ and DOI: https://doi.org/10.17605/OSF.IO/MKXQE.

The following datasets were generated:

| Author(s) | Year | Dataset title | Dataset URL | Database and Identifier |
|---|---|---|---|---|
| Tina B Lonsdorf, Maren Klingelhöfer-Jens | 2019 | Data_and_code_dataset1 | https://osf.io/w9y8z/ | Open Science Framework, w9y8z |
| Tina B Lonsdorf, Rachel Sjouwerman | 2019 | Data_and_code_dataset2 | https://osf.io/7c5ag/ | Open Science Framework, 7c5ag |

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

**Appendix 1—table 1.** Summary of criteria used to define 'non-learners' across records included in the systematic literature search. Criteria used to define 'non-learners' were identified in eleven records reported in a total of 14 individual studies.

| Reference | % excluded participants ('non-learners') | CS+/CS– cut-off (in µS) for 'non-learners' | N trials_acq total CS+/CS– | N trials_acq considered CS+/CS– | Trials_phase (unless otherwise stated, this refers to fear acquisition training) | Additional criteria/notes |
|---|---|---|---|---|---|---|
| Ahmed and Lovibond, 2019, Exp. 1 | 24% | <or = 0 | 3/3 | 2/2 | last two thirds | only considered as 'non-learners' if applicable to both SCL and ratings |
| Ahmed and Lovibond, 2019, Exp. 2 | 16% | | | | | |
| Reddan et al., 2018[a] | 35% | <or = 0 | 16/8 | 8[b]/8 | full phase | |
| Grégoire and Greening, 2019 | 16% | <0.1 | 13[c]/8 | 4[c]/4 | last third | participants were also excluded if they did not show equivalent responding to both CS+s (difference > 0.1 µS) or when not showing equal extinction to both CS+s or complete differential extinction to both CS+s vs. CS– (difference > 0.1 µS) |
| Hu et al., 2018 | 6% | <or = 0 | 16/10 | 5/5[d] OR 1/1 | second half[d] OR last trial[d] | 'non-learners' discontinued after day 1 of the experiment |
| Oyarzún et al., 2019, Exp. 1 | 27% | <or = 0[e] | eight[c]/8 | 4[c]/4 | second half | only considered as 'non-learners' if applicable to both SCR and fear-potentiated startle |
| Oyarzún et al., 2019, Exp. 2 | 41% | | | | | |
| Belleau et al., 2018 | 2% | <0.05 | 5/5 | 5/5 | full phase | only considered as 'non-learners' when also failing to show any differential ratings (i.e.,<or = 0 in discrimination)[f] |
| Morriss et al., 2018 | 6% g | < or = 0[h] | 12/6 | 6[b]/6 | full phase | only considered as 'non-learners' if applicable across all phases (fear acquisition and extinction training, avoidance acquisition and extinction) |

*Appendix 1—table 1 continued*

| Reference | % excluded participants ('non-learners') | CS+/CS- cut-off (in µS) for 'non-learners' | N trials$_{acq}$ total CS+/CS- | N trials$_{acq}$ considered CS+/CS- | Trials $_{phase}$ (unless otherwise stated, this refers to fear acquisition training) | Additional criteria/notes |
|---|---|---|---|---|---|---|
| **Schiller et al., 2018; Schiller et al., 2010, Exp. 1** | 48%[i] | <0.1/mean SCR to the US | 16[c]/10, 13[c]/8 | 5[b]/5 OR 5[b]/5 OR 1[b]/1 OR increase from first to last trial | first half of acquisition OR second half OR last trial of acquisition, OR the increase from the first to last trial of acquisition | |
| **Schiller et al., 2018; Schiller et al., 2010, Exp. 2** | 74%[i] | | | 4[b]/4 OR 4[b]/4 OR 1[b]/1 OR increase from first to last trial | | |
| **Nitta et al., 2018** | 52% | <0 | 13[c]/8 | 2[c]/2 | last two trials | one additional participant showed strong SCR during re-extinction phase to CS+ and was therefore excluded |
| **Hartley et al., 2019** | 16% | <or = 0.05 | 6/6 | 3/3 | last half | |
| **Hu et al., 2019** | 16% | < 0[k] | 8[b]/8 | 4[b]/4 | second half | |

[a]Personal communication with D. Schiller (20.5.2019 and 30.8.2019) confirmed that individuals were classified as 'non-learners' when they 'did not demonstrate greater SCRs to the CS+ relative to the CS- on average across all acquisition trials (n = 24)' (see 'Materials and methods' section). The personal communication clarified that the statement included in the results section that defines 'non-learners' as individuals that "did not demonstrate a discriminatory SCR during acquisition, defined as greater SCR to the CS+ relative to the CS- during either the first or last half of threat-acquisition on average' was intended to refer to the same procedure (i.e., the full acquisition phase).

[b]Refers to unreinforced CS+ trials (CS+ trials not followed by the US) only.

[c]For each CS+1 and CS+2.

[d]Personal communication with D. Schiller (1.5.2019): late acquisition as reported in the publication refers to the last half or last trial. Anyone that had a positive difference (>0.000 µS) in either the second half or last trial of acquisition was kept.

[e]Personal communication with J. Oyarzun (21.5.2019): all difference scores > 0 µS were considered as CS+/CS- discrimination.

[f]Personal communication with E. Balleau, PhD (5.5.2019): 'differential ratings' means that CS+>CS is equal to or below 0 µS was non-differentiation.

[g]These were not excluded as results did not change.

[h]Personal communication with J. Morriss (15.4.2019): no positive differential response is defined as any number <0 µS.

[i]Percentages for 'non-learners', 'non-extinguishers' and 'non-responders' reported together.

[k]Personal communication with D. Schiller (21.5.2019): zero differences were kept.

# Appendix 1

## Definition of performance-based exclusion of participants ('non-learners') and numbers of participants excluded across studies

**Appendix 1—table 2. Summary of criteria used to define 'non-responders' across records included in the systematic literature search.** Fifteen records, reporting a total of 17 studies, were identified.

| Record | % excluded participants ('non-responders') | Cut-off (in $\mu S$) for a valid SCR | Valid responses in at least % of trials | Stimulus type (also referred to as 'trial') on which the exclusion is based | Additional criteria/notes |
|---|---|---|---|---|---|
| *Baeuchl et al., 2019* | 10% | >0.01 | ≥66% | CXT+ | |
| *Tuominen et al., 2019* | 12% | >0.05 | ≥13% | CS+ and CS– | |
| *Gruss and Keil, 2019* | 11% | visual inspection[a] | | CS+, CS–and US | |
| *Sjouwerman and Lonsdorf, 2019* | 14% | ≥0.02 | US:≥67% CS: no valid response in each CS modality | CS+, CS– and US | |
| *Grégoire and Greening, 2019* | 8% | >0.02 | ≥25%[b] | CS+ and CS–[c] | |
| *Hu et al., 2018* | 3% | ≥0.02 | 100% | CS+ and CS–[c] | 'non-responders' discontinued after day 1 of the experiment |
| *Oyarzún et al., 2019*, Exp. 1 | 0% | ≥0.02 | ≥25% | CS+[d] and CS–[c] | |
| *Oyarzún et al., 2019*, Exp. 2 | 9% | | | | |
| *Tani et al., 2019* | 10% | >0.03[e] | 100% | CS+ | |
| *Marin et al., 2019* | 0%[f] | ≥0.03 | ≥10% | US | |
| *Taylor et al., 2018* | 5% | NA | 100% | motor test[g] | |
| *Morriss et al., 2018* | 6% | >0.03 | ≥90% | CS+[d] and CS–[c] | only applicable if true across all phases/days of the experiment |
| *Schiller et al., 2018*; *Schiller et al., 2010*, Exp. 1 | 48%[h] | ≥0.02 | ≥25% | CS+[d] and CS– | |
| *Schiller et al., 2018*; *Schiller et al., 2010*, Exp 2 | 74%[h] | | | | |

*Appendix 1—table 2 continued on next page*

*Appendix 1—table 2 continued*

| Record | % excluded participants ('non-responders') | Cut-off (in μS) for a valid SCR | Valid responses in at least % of trials | Stimulus type (also referred to as 'trial') on which the exclusion is based | Additional criteria/notes |
|---|---|---|---|---|---|
| *Morriss and van Reekum, 2019*, Exp. 1 | 2% | >0.03 | >90% | CS+[d] and CS– | |
| *Morriss and van Reekum, 2019*, Exp. 2 | 2% | | | | |
| *Hartley et al., 2019* | 6% | <0.05[i] | ≥33 %[i] | CS+ and CS–[c] | |
| *Hu et al., 2019* | 4% [k] | ≥0.02 | 100% [k] | US | |
| *Leuchs et al., 2019* | 4% | NA | ≥33% | CS+ and CS–[c] | only applicable if true across both days of the experiment |

[a]Personal communication with L. Forest Gruss (29.4.2019): "the determination of non-responders was done, this was done on visual inspection by me through all trials of all individuals. I verified after determining who the lowest, i.e. non-responders were, in the same fashion as the startle non-responders in summing responding over the entire experiment, and this responding falling below a threshold of overall response (~<10%) AND one individual due to lack of response at the end of the trial to the UCS specifically".

[b]Personal communication with S.G. Greening (24.4.2019): "non-responders if more than 75% of data were missing (i.e., SCR <0.02 μS) during the training phase. So, that means, if a participant had at least six trials (out of 24) with measurable SCRs (whatever the condition), we kept them (if the other acquisition criteria were OK, see below). If they had five trials or fewer with measurable GSR, we considered them a non-responder and removed them".

[c]Personal communications that 'trial' or this statement refers to CS+ and CS– trials: S. Greening (24.4.2019), D. Schiller (1.5.2019), J. Oyarzun (21.5.2019), J. Morriss (15.4.2019), C. Hartley (2.5.2019), V. Spoormaker (18.4.2019).

[d] CS+ unpaired.

[e]Personal communication with H. Tani (2.5.2019): only CS+ trials were considered (here as response to the sound or the intrapersonal stimulus).

[f]Personal communication with M.-F. Marin (23.4.2019): exclusion criteria were defined, but no participant met these criteria and hence none was excluded.

[g]Personal communication with V. Taylor (6.6.2019): clarified that "non-responders' were identified in a "motor test of SCR responding during the preliminary session. Essentially, they had to compress a ball with the right hand with maximal physical force for a few seconds on about 10 trials, which typically elicits quite large SCRs in subjects. Failure to respond to an SCR to all of these trials was considered a non-responder".

[h]Percentages for 'non-learners', 'non-extinguishers' and 'non-responders' reported together.

[i]Personal communication with C. Hartley (2.5.2019): clarified that "participants were considered non-responder if they had SCR values of 0 for more than 8 of the 12 trials in acquisition (<4 responsive trials)".

[k] The percentage of 'non-responders' and 'non-learners' was reported together without percentages for each category; personal communication with D. Schiller (21.5.2019): in the paper, it is reported that five individuals 'were excluded due to equipment malfunction (N = 2) or had non-measurable skin conductance response (SCR) to the shock (N = 3)". It was confirmed that these individuals excluded for non-measurable SCR did not show any responses to any stimulus.

## Appendix 2

# Applying the identified performance-based exclusion criteria to existing data: a case example

In this case example based on Data set 1 (see main manuscript), we tested whether CS+/CS– discrimination in SCRs does indeed differ between the different exclusion groups as defined by the cut-offs retrieved from the literature (see *Figure 2B*). Note that this is somewhat circular as exclusion groups are defined by different SCR CS+/CS– cutoffs, which then are used in an analysis in which differential SCRs are the dependent measure. However, that this is exactly what is sometimes done in the literature (see main manuscript).

Still, this is an important manipulation check to test empirically whether those classified in a group of 'non-learners' in the literature do indeed show no evidence of learning, which would be indicated by comparable SCRs to the CS+ and the CS– (i.e., no significant discrimination). Here, we test this for cumulative exclusion groups. Note that this is only a rough manipulation check, as a non-significant CS+/CS– discrimination effect in the whole group (e.g., those showing a CS+/CS– discrimination <0.05 µS based on raw scores) cannot be taken as evidence that all individuals in this group do not display meaningful or statistically significant CS+/CS– discrimination. More precisely, half of this group who did not meet the cut-off of 0.05µS in CS+/CS– discrimination do show a negative or zero discrimination score, which may bias the group average score towards non-discrimination. Yet, statistically testing for discrimination within each exclusion group (e.g. specifically in the group showing a discrimination between >0 and < 0.05 µS) is not unproblematic.

**Appendix 2—table 1.** Results of two-tailed t-tests for differences in SCR CS+/CS– discrimination in Data set 1 for the different cumulative exclusion groups (indicated by the + in the table) based on the criteria identified in the literature with respect to CS+/CS– discrimination cutoffs (in µS). For completeness sake and as it is not always clear whether CS+/CS– discrimination is based on raw or transformed values, we report results based on analyses of both raw (A) and transformed values (B). P-values for these post-hoc tests are Bonferroni corrected.

**A) t-tests: CS+/CS– discrimination based on raw values**

| Exclusion group (cumulative) | CS+ M (SD) | CS– M (SD) | df | t | p_bonf_corr | d |
|---|---|---|---|---|---|---|
| <0 | 0.04 (0.04) | 0.07 (0.07) | 10 | −2.67 | .140 | 0.81 |
| + = 0 | 0.02 (0.04) | 0.03 (0.05) | 33 | −2.24 | .193 | 0.38 |
| + > 0 and < 0.05 | 0.04 (0.05) | 0.03 (0.05) | 66 | 2.14 | .219 | 0.26 |
| + = 0.05 | 0.04 (0.05) | 0.03 (0.05) | 70 | 2.88 | .031 | 0.34 |
| + > 0.05 and < 0.1 | 0.06 (0.06) | 0.04 (0.05) | 88 | 5.87 | .0000005 | 0.62 |
| + ≥ 0.1 | 0.10 (0.10) | 0.04 (0.06) | 115 | 7.87 | <0.000000001 | 0.73 |

**B) t-tests: CS+/CS– discrimination based on log-transformed and range-corrected values**

| Exclusion group (cumulative) | CS+ M (SD) | CS– M (SD) | df | t | p_bonf_corr | d |
|---|---|---|---|---|---|---|
| <0 | 0.09 (0.10) | 0.13 (0.11) | 13 | −3.46 | 0.025 | 0.93 |
| + = 0 | 0.04 (0.08) | 0.06 (0.10) | 28 | −2.90 | 0.043 | 0.54 |
| + > 0 and < 0.05 | 0.06 (0.10) | 0.07 (0.11) | 42 | −0.88 | >0.999 | 0.13 |
| + = 0.05 | 0.07 (0.10) | 0.07 (0.11) | 46 | −0.06 | >0.999 | 0.01 |
| + > 0.05 and < 0.1 | 0.09 (0.11) | 0.07 (0.11) | 60 | 2.81 | .040 | 0.36 |
| + ≥ 0.1 | 0.21 (0.19) | 0.10 (0.11) | 115 | 9.56 | <0.000000001 | 0.89 |

## Appendix 3

### Exploratory analyses on consistency of classification ('learners' vs. 'non-learners') across outcome measures and criteria employed

Throughout the main manuscript and particularly in the discussion, we highlight that differential (CS+>CS–) SCRs alone cannot be taken to infer 'learning' (*Figure 4—figure supplement 1*).

*Appendix 3—table 1* provides statistical information on CS+/CS– discrimination in fear ratings in (cumulative) exclusion groups as defined by CS+/CS– discrimination in SCRs.

**Appendix 3—table 1.** CS+/CS– discrimination in fear ratings in (cumulative) exclusion groups (indicated by the + in the table) as defined by CS+/CS– discrimination in SCRs (based on raw scores).

| Exclusion group (cumulative) | CS+ M (SD) | CS– M (SD) | df | t | $p_{bonf\_corr}$ | d |
|---|---|---|---|---|---|---|
| <0 | 15.8 (8.94) | 2.45 (4.70) | 10 | 5.37 | 0.002 | 1.62 |
| + = 0 | 16.6 (7.73) | 3.15 (5.82) | 31 | 9.69 | <0.000000001 | 1.71 |
| + > 0 and < 0.05 | 16.2 (7.37) | 3.06 (5.86) | 64 | 12.8 | <0.000000001 | 1.59 |
| + = 0.05 | 16.3 (7.26) | 2.96 (5.75) | 67 | 13.4 | <0.000000001 | 1.62 |
| + > 0.05 and < 0.1 | 16.5 (6.97) | 2.94 (5.47) | 84 | 16.0 | <0.000000001 | 1.74 |
| + >= 0.1 | 17.3 (6.64) | 3.08 (5.04) | 110 | 20.2 | <0.000000001 | 1.92 |

**Appendix 4—table 1.** Overview of SCR response quantification specifications (i.e., min. amplitude, scoring approach) and procedural details during fear acquisition training (i.e., number of CS and US presentations) as well as number (mean and range) and percentage of SCR non-responses towards the different stimuli (US, CS+, CS−, CS).

TTP: trough-to-peak; CS+E: CS+ extinguished; CS+U: CS+ unextinguished, CS: for both the CS+ and CS−.

| Reference | N | Minimum amplitude cutoff (in μS) for valid SCRs | Scoring details | Number of... US | Number of... CS (CS+/CS−) | Non-responses towards... US (M ± SD, range) | US (%) | CS+ (M ± SD, range) | CS+ (%) | CS− (M ± SD, range) | CS− (%) | CS (M ± SD, range) | CS (%) |
|---|---|---|---|---|---|---|---|---|---|---|---|---|---|
| Jentsch et al., 2020 | 41 | ≥0.02 | TTP (max peak), latency 0.5-4 s/ 1-80.5 s (US/CS) | 10 | 16/16 | 1.12 ± 1.66 (0-10) | 11.22 | 2.22 ± 3.31 (0-16) | 13.87 | 4.49 ± 3.92 (0-16) | 28.05 | 6.71 ± 6.68 (0-32) | 20.96 |
| Hermann et al., 2016 | 45 | ≥0.02 | TTP (max peak), latency 0.5-6 s/ 1-60.5 s (US/CS) | 10 (5 for CS+E, 5 for CS+U) | 8 CS+E/8 CS+U/16 CS− | 0.24 ± 0.88 (0-5) | 2.44 | 2.64 ± 3.49 (0-13); CS+E: 1.47 ± 2.19 (0-8); CS+U: 1.18 ± 1.80 (0-7) | 16.53 CS+E: 18.33; CS+U: 14.72 | 8.07 ± 4.14 (0-16) | 50.42 | 10.71 ± 6.65 (0-26) | 33.47 |
| Merz et al., 2018a | 39 | ≥0.02 | TTP (max peak), latency 0.5-6 s/ 1-60.5 s (US/CS) | 10 (5 for CS+E, 5 for CS+U) | 8 CS+E/8 CS+U/8 CS− | 2.08 ± 1.98 (0-8) | 20.77 | 3.36 ± 4.55 (0-16); CS+E: 1.59 ± 2.35 (0-8); CS+U: 1.77 ± 2.32 (0-8) | 21.00; CS+E: 19.87; CS+U: 22.12 | 2.41 ± 2.27 (0-8) | 30.13 | 5.77 ± 6.49 (0-24) | 24.04 |
| Merz et al., 2014 | 40 | ≥0.02 | TTP (max peak), latency 0.5-6 s/ 1-60.5 s (US/CS) | 10 (5 for CS+E, 5 for CS+U) | 8 CS+E/8 CS+U/16 CS− | 0.13 ± 0.33 (0-1) | 1.25 | 1.08 ± 2.04 (0-11); CS+E: 0.58 ± 1.08 (0-5); CS+U: 0.50 ± 1.11 (0-6) | 6.72; CS+E: 7.19; CS+U: 6.25 | 3.13 ± 2.96 (0-11) | 19.53 | 4.20 ± 4.39 (0-21) | 13.13 |
| Hamacher-Dang et al., 2015 | 39 | ≥0.02 | TTP (max peak), latency 0.5-6 s/ 1-60.5 s (US/CS) | 10 (5 for CS+E, 5 for CS+U) | 8 CS+E/8 CS+U/16 CS− | 0.23 ± 0.48 (0-2) | 2.31 | 2.33 ± 3.77 (0-12); CS+E: 1.31 ± 2.21 (0-8); CS+U: 1.03 ± 1.81 (0-7) | 14.58; CS+E: 16.35; CS+U: 12.82 | 3.77 ± 4.20 (0-14) | 23.56 | 6.10 ± 7.71 (0-26) | 19.07 |
| Mertens et al., 2019 | 59 | ≥0.02 | TTP (max peak), latency 1-8 s (baseline 0-2 s) | 10 | 10/5 | 0.78 ± 1.69 (0-6) | 7.8 | 4.75 ± 2.97 (0-10) | 47.5 | 2.93 ± 1.66 (0-5) | 58.6 | 7.68 ± 4.30 (0-15) | 51.2 |
| Klingel-höfer-Jens et al., unpublished | 119 | ≥0.01 | TTP (first peak), latency 0.9- 2.5 s/ 3.5 s (US/CS) | 14 | 14/14 | 1.40 ± 2.47 (0-14) | 10.0 | 5.30 ± 4.42 (0-14) | 37.9 | 8.20 ± 3.99 (0-14) | 58.6 | 6.75 ± 4.44 (0-14) | 48.2 |
| Gerlicher et al. unpublished | 52 | ≥0.02 | TTP (first peak) latency 0.9-4 s | 6 | 6/6 | 0.73 ± 1.39 (0-6) | 12.18 | 2.73 ± 2.06 (0-6) | 45.5 | 3.54 ± 1.82 (0-6) | 59.0 | 6.27 ± 3.54 (0-12) | 52.24 |

*Appendix 4—table 1 continued on next page*

## Appendix 4—table 1 continued

| Reference | N | Minimum amplitude cutoff (in μS) for valid SCRs | Scoring details | Number of... US | CS (CS+/CS−) | 'Non-responses' towards... US (M ± SD, range) | US (%) | CS+ (M ± SD, range) | CS+ (%) | CS− (M ± SD, range) | CS− (%) | CS (M ± SD, range) | CS (%) |
|---|---|---|---|---|---|---|---|---|---|---|---|---|---|
| Gerlicher et al., 2018 | 39 | ≥0.02 | TTP (first peak) latency 0.9–4 s | 5 | 10/10 | 0.33 ± 0.93 (0–5) | 6.67 | 1.05 ± 2.21 (0–10) | 10.51 | 2.36 ± 2.49 (0–10) | 23.59 | 3.41 ± 4.48 (0–20) | 17.05 |
| Andreatta et al. unpublished | 76 | ≥0.02 | TTP (first peak) latency 0.8–4 s | 16 (8 in analysis due to startle probes) | 16/16 (8/8 in analysis due to startle probes) | 1.34 ± 1.69 (0–8) | 16.78 | 4.17 ± 2.30 (0–8) | 52.14 | 5.00 ± 1.98 (0–8) | 62.50 | 9.17 ± 3.77 (0–16) | 57.32 |
| Wendt et al., 2020 | 112 | ≥0.04 | TTP (first peak), latency 0.9–4 s | 9 | 12/12 | 0.46 ± 1.15 (0–7) | 5.06 | 5.88 ± 3.63 (0–12) | 48.96 | 7.06 ± 3.19 (0–12) | 58.85 | 12.94 ± 6.39 (0–24) | 53.91 |
| Wendt et al., 2015 | 108 | ≥0.04 | TTP (first peak), latency 0.9–4 s | 12 | 12/12 | 0.27 ± 0.99 (0–8) | 2.24 | 6.44 ± 3.81 (0–12) | 53.63 | 8.53 ± 2.65 (0–12) | 71.06 | 14.96 ± 6.04 (0–24) | 62.35 |
| Drexler et al., 2015 | 46 | ≥0.02 | TTP (max peak), latency 1–4.5 s | 18 | 13 CS1+/13 CS2+/13 CS− | 2.8 ± 4.18 (0–16) | 15.57 | 9.67 ± 7.64 (0–26); CS1+: 4.87 ± 4.07 (0–13); CS2+: 4.80 ± 3.78 (0–13) | 37.20; CS1+:37.45; CS2+: 36.95 | 5.26 ± 3.95 (0–13) | 40.46 | 14.93 ± 11.37 (0–39) | 38.29 |
| Meir Drexler et al., 2016 | 73 | ≥0.02 | TTP (max peak), latency 1–4.5 s | 18 | 13 CS1+/13 CS2+/13 CS− | 3.37 ± 4.72 (0–18) | 18.72 | 11.51 ± 7.96 (0–25); CS1+: 5.78 ± 3.97 (0–13); CS2+: 5.73 ± 4.22 (0–13) | 44.25; CS1+:44.67; CS2+:44.04 | 6.29 ± 3.94 (0–13) | 48.36 | 17.79 ± 11.67 (0–37) | 45.62 |
| Meir Drexler and Wolf, 2017 | 72 | ≥0.02 | TTP (max peak), latency 1–4.5 s | 18 | 13 CS1+/13 CS2+/13 CS− | 1.92 ± 2.96 (0–11) | 10.64 | 9.65 ± 7.21 (0–25); CS1+: 4.78 ± 3.72 (0–12); CS2+: 4.88 ± 3.85 (0–13) | 37.12; CS1 +: 36.75; CS2+: 37.50 | 5.42 ± 3.54 (0–12) | 41.66 | 15.07 ± 10.40 (0–36) | 38.63 |
| Drexler et al., 2018 | 40 | ≥0.02 | TTP (max peak), latency 1–4.5 s | 10 (5 for CS+E, 5 for CS+U) | 8 CS+E/8 CS+U/16 CS− | 0.32 ± 0.69 (0–3) | 3.25 | 4.17 ± 4.45 (0–16); CS+E: 2.02 ± 2.47 (0–8); CS+U: 2.15 ± 2.38 (0–8) | 26.09; CS +E: 25.31; CS+U: 26.87 | 6.07 ± 4.37 (0–16) | 37.96 | 10.25 ± 8.24 (1–27) | 32.03 |
| Meir Drexler et al., 2019 | 75 | ≥0.02 | TTP (max peak), latency 0.5–6 s/ 1–80.5 s (US/CS) | 6 | 10/10 | 0.89 ± 01.57 (0–6) | 14.88 | 4.07 ± 3.40 (0–10) | 40.66 | 4.68 ± 3.23 (0–10) | 46.8 | 8.75 ± 6.41 (0–20) | 43.73 |

Appendix 4—table 1 continued on next page

*Appendix 4—table 1 continued*

| Reference | N | Minimum amplitude cutoff (in μS) for valid SCRs | Scoring details | Number of... US | Number of... CS (CS+/CS−) | 'Non-responses' towards... US (M ± SD, range) | US (%) | CS+ (M ± SD, range) | CS+ (%) | CS− (M ± SD, range) | CS− (%) | CS (M ± SD, range) | CS (%) |
|---|---|---|---|---|---|---|---|---|---|---|---|---|---|
| Chalkia et al., unpublished | 238 | ≥0.02 | TTP (first peak), latency 0.5–4.5 s | 6 | 16/10 (10/10 in analysis, only unreinforced trials) | 0 (0–6) | 0 | 0.03 ± 0.19 (0–10) | 0.29 | 0.05 ± 0.29 (0–10) | 0.50 | 0.08 ± 0.42 (0–20) | 0.40 |
| Hollandt et al., unpublished | 30 | >0.04 | TTP (first peak), latency 0.9–4 s | 6 | 10/10 | 0 | 0 | 2.97 ± 2.81 (0–10) | 29.67 | 7.23 ± 2.61 (0–10) | 72.33 | 10.20 ± 4.72 | 51.0 |
| *Sjouwerman et al., 2018* | 326 | ≥0.02 | TTP (first peak), latency 0.9–4.5 s | 9 | 9/9 | 1.38 ± 1.73 (0–9) | 15.37 | 3.11 ± 2.69 (0–9) | 34.59 | 3.77 ± 2.68 (0–9) | 41.92 | 6.87 ± 5.01 (0–18) | 38.26 |

## Appendix 4

### Definition of 'non-responders 'and amount of participants excluded across studies

In the main manuscript, we discuss different frequencies of 'non-responding' to different experimental stimuli (e.g., US, CS+ and CS– in isolation or in combination), which inherently lead to different exclusion frequencies when classifying 'non-responders' on the basis of different types of stimuli. As there is little empirical work on the frequency of 'non-responses' to the US, CSs (i.e., CS+ and CS–) and CS+ only to base recommendations on, we compiled this information across 20 different data sets (see *Appendix 4—table 1*), including information on SCR response quantification specifications (i.e., minimum amplitude, scoring approach) and procedural details during fear acquisition training (i.e., number of CS and US presentations). These data sets were provided by different co-authors involved in this manuscript.

In addition, *Appendix 4—table 2* provides information on the number and percentage of individuals in a sample showing SCR 'non-responses' to a certain number of US presentations during fear acquisition training as well as mean number and percentage of CS responses (CS refers to the CS+ and CS– combined) in these individuals to guide the development of empirically based criteria to define SCR 'non-responders'.

**Appendix 4—table 2.** Number and percentage of individuals in a sample showing SCR non-responses to a certain number of US presentations during fear acquisition training (exemplarily for one to eight USs[#]), as well as mean number of and percentage of CS responses (CS refers to the CS+ and CS– combined) in these individuals. [#]Here only up to eight USs are included as eight is half of the maximum number of US presentations in the samples included here.

| Reference | a) *n* (%) of individuals with 0, 1, 2, 3, 4, 5, 6, 7, and 8 SCRs towards the US. b) *M* (%) of valid CS responses for these individuals. | | | | | | | | |
|---|---|---|---|---|---|---|---|---|---|
| | 0 US | 1 US | 2 US | 3 US | 4 US | 5 US | 6 US | 7 US | 8 US |
| *Jentsch et al., 2020* | a) 1 (2.4%) b) 0 (0%) | a) 0 (0%) b) NA | a) 0 (0%) b) NA | a) 0 (0%) b) NA | a) 0 (0%) b) NA | a) 0 (0%) b) NA | a) 0 (0%) b) NA | a) 2 (4.9%) b) 27.5 (85.9%) | a) 7 (17.1%) b) 25.4 (79.5%) |
| *Hermann et al., 2016* | a) 0 (0%) b) NA | a) 0 (0%) b) NA | a) 0 (0%) b) NA | a) 0 (0%) b) NA | a) 0 (0%) b) NA | a) 1 (2%) b) 12 (37.5%) | a) 0 (0%) b) NA | a) 1 (2%) b) 14 (43.7%) | a) 0 (0%) b) NA |
| *Merz et al., 2018a* | a) 0 (0%) b) NA | a) 0 (0%) b) NA | a) 1 (2.6%) b) 23.0 (95.8%) | a) 0 (0%) b) NA | a) 2 (5.1%) b) 20.0 (83.3%) | a) 3 (7.7%) b) 23.0 (95.8%) | a) 1 (2.6%) b) 21.0 (87.5%) | a) 5 (12.8%) b) 21.4 (89.1%) | a) 9 (23.1%) b) 21.6 (85.6%) |
| *Merz et al., 2014* | a) 0 (0%) b) NA | a) 0 (0%) b) NA | a) 0 (0%) b) NA | a) 0 (0%) b) NA | a) 0 (0%) b) NA | a) 0 (0%) b) NA | a) 0 (0%) b) NA | a) 0 (0%) b) NA | a) 0 (0%) b) NA |
| *Hamacher-Dang et al., 2015* | a) 0 (0%) b) NA | a) 0 (0%) b) NA | a) 0 (0%) b) NA | a) 0 (0%) b) NA | a) 0 (0%) b) NA | a) 0 (0%) b) NA | a) 0 (0%) b) NA | a) 0 (0%) b) NA | a) 1 (3%) b) 24 (75.0%) |
| *Mertens et al., 2019* | a) 0 (0%) b) NA | a) 0 (0%) b) NA | a) 0 (0%) b) NA | a) 0 (0%) b) NA | a) 4 (6.78%) b) 1.75 (11.67%) | a) 0 (0%) b) NA | a) 2 (3.39%) b) 3.5 (23.33%) | a) 2 (3.39%) b) 9 (60%) | a) 0 (0%) b) NA |
| Klingelhöfer-Jens et al., unpublished | a) 2 (1.68%) b) 0 (0%) | a) 0 (0%) b) NA | a) 1 (0.84%) b) 10 (35.7%) | a) 0 (0%) b) NA | a) 0 (0%) b) NA | a) 0 (0%) b) NA | a) 1 (0.84%) b) 1 (3.57%) | a) 2 (1.68%) b) 2 (7.14%) | a) 1 (0.84%) b) 0 (0%) |

*Appendix 4—table 2 continued on next page*

*Appendix 4—table 2 continued*

| | a) *n* (%) of individuals with 0, 1, 2, 3, 4, 5, 6, 7, and 8 SCRs towards the US. b) *M* (%) of valid CS responses for these individuals. | | | | | | | | |
|---|---|---|---|---|---|---|---|---|---|
| Reference | 0 US | 1 US | 2 US | 3 US | 4 US | 5 US | 6 US | 7 US | 8 US |
| Gerlicher et al., unpublished | a) 0 (0%) b) NA | a) 0 (0%) b) NA | a) 0 (0%) b) NA | a) 3 (5.77%) b) 4 (33.33%) | a) 5 (9.62%) b) 4.8 (40%) | a) 7 (13.46%) b) 6.7 (55.91%) | a) 35 (67.31%) b) 6.15 (51.25%) | NA | NA |
| *Gerlicher et al., 2018* | a) 1 (2.56%) b) 0 (0%) | a) 0 (0%) b) NA | a) 0 (0%) b) NA | a) 2 (5.13%) b) 19.5 (97.5%) | a) 4 (10.26%) b) 17.5 (87.50%) | a) 32 (82.05%) b) 16.81 (84.05%) | NA | NA | NA |
| *Wendt et al., 2020* | a) 0 (0%) b) NA | a) 0 (0%) b) NA | a) 1 (0.9%) b) 18 (75%) | a) 1 (0.9%) b) 24 (100%) | a) 1 (0.9%) b) 0 (0%) | a) 0 (0%) b) NA | a) 2 (1.8%) b) 12 (50%) | a) 8 (7.1%) b) 13.13 (54.69%) | a) 11 (9.9%) b) 11.09 (46.21%) |
| *Wendt et al., 2015* | a) 0 (0%) b) NA | a) 0 (0%) b) NA | a) 0 (0%) b) NA | a) 0 (0%) b) NA | a) 1 (0.9%) b) 18 (75%) | a) 0 (0%) b) NA | a) 0 (0%) b) NA | a) 1 (0.9%) b) 17 (70.83%) | a) 0 (0%) b) NA |
| *Drexler et al., 2015* | a) 0 (0%) b) NA | a) 0 (0%) b) NA | a) 2 (4.3%) b) 0.5 (1.28%) | a) 1 (2.2%) b) 0.0 (0.0%) | a) 0 (0%) b) NA | a) 0 (0%) b) NA | a) 0 (0%) b) NA | a) 1 (2.2%) b) 7.0 (17.94%) | a) 0 (0%) b) NA |
| *Meir Drexler et al., 2016* | a) 1 (1.4%) b) 29.00 (74.35%) | a) 0 (0%) b) NA | a) 2 (2.7%) b) 2.0 (5.12%) | a) 1 (1.4%) b) 9.0 (23.07%) | a) 1 (1.4%) b) 2.0 (5.12%) | a) 1 (1.4%) b) 3.0 (7.69%) | a) 0 (0%) b) NA | a) 4 (5.5%) b) 5.0 (12.82%) | a) 2 (2.7%) b) 6.50 (16.66%) |
| *Meir Drexler and Wolf, 2017* | a) 0 (0%) b) NA | a) 0 (0%) b) NA | a) 0 (0%) b) NA | a) 0 (0%) b) NA | a) 0 (0%) b) NA | a) 0 (0%) b) NA | a) 0 (0%) b) NA | a) 1 (1.4%) b) 5.0 (12.82%) | a) 1 (1.4%) b) 5.0 (12.82%) |
| *Drexler et al., 2018* | a) 0 (0%) b) NA | a) 0 (0%) b) NA | a) 0 (0%) b) NA | a) 0 (0%) b) NA | a) 0 (0%) b) NA | a) 0 (0%) b) NA | a) 0 (0%) b) NA | a) 1 (2.5%) b) 8 (25%) | a) 2 (5.0%) b) 12.5 (39.06%) |
| *Meir Drexler et al., 2019* | a) 3 (4.0%) b) 0.33 (1.66%) | a) 1 (1.3%) b) 1 (5.0%) | a) 4 (5.3%) b) 4.25 (21.25%) | a) 2 (2.7%) b) 3.0 (15.0%) | a) 3 (4.0%) b) 1.33 (6.66%) | a) 19 (25.3%) b) 12.63 (63.15%) | a) 43 (57.3%) b) 13.21 (66.04%) | a) 0 (0%) b) NA | a) 0 (0%) b) NA |
| Chalkia et al., unpublished | a) 0 (0%) b) NA | a) 0 (0%) b) NA | a) 0 (0%) b) NA | a) 0 (0%) b) NA | a) 0 (0%) b) NA | a) 0 (0%) b) NA | a) 238 (100%) b) 19.92 (99.6%) | a) 0 (0%) b) NA | a) 0 (0%) b) NA |
| Hollandt et al., unpublished | a) 0 (0%) b) NA | a) 0 (0%) b) NA | a) 0 (0%) b) NA | a) 0 (0%) b) NA | a) 0 (0%) b) NA | a) 0 (0%) b) NA | a) 0 (0%) b) NA | NA | NA |
| *Sjouwerman et al., 2018* | a) 4 (1.23%) b) 0.5 (2.78%) | a) 2 (0.61%) b) 2.5 (13.89%) | a) 4 (1.23%) b) 4.13 (22.92%) | a) 2 (0.61%) b) 7.25 (40.28%) | a) 0 (0%) b) NA | a) 0 (0%) b) NA | a) 0 (0%) b) NA | a) 0 (0%) b) NA | a) 0 (0%) b) NA |

