## [Decision Letter]

**Acceptance summary:**

Fear is a central dimension of childhood temperament and adult personality. When extreme, fear contributes to the development a spectrum of prevalent and debilitating psychiatric disorders. Mechanistic studies in mice and rats have begun to illuminate the specific molecules, cells, and circuits that control the expression of fear-relevant behaviors, but the relevance of these tantalizing discoveries to the complexities of the human brain and human suffering remain unclear. Human fear conditioning studies represent a crucial means of the addressing this question, yet as Lonsdorf and colleagues make clear in this report, the impact of such studies is limited by several crucial (but seemingly mundane) methodological issues. In particular, they highlight substantial heterogeneity in exclusion criteria for such studies-who does and who does not adequately learn the association between cues and consequences. Based on this, they propose evidence-based clear guidelines for conducting and reporting fear conditioning studies. This report is likely to be of keen interest to a broad spectrum of stakeholders, including the producers, consumers, and funders of human fear research.

**Decision letter after peer review:**

Thank you for submitting your article "Lessons from fear conditioning on exclusions in learning tasks on how to not get lost in the garden of forking paths" for consideration by *eLife*. Your article has been reviewed by three peer reviewers, and the evaluation has been overseen by Dr. Shackman as the Reviewing Editor and Dr. de Lange as the Senior Editor. The following individuals involved in review of your submission have agreed to reveal their identity: Daniel Bradford (Reviewer #1); Christine Larson (Reviewer #2); Nick Balderston (Reviewer #3).

The reviewers have discussed the reviews with one another and the Reviewing Editor has drafted this decision to help you prepare a revised submission.

As the affective neuroscientist Kay Tye recently noted, "Fear resembles a dictator that makes all other brain processes (from cognition to breathing) its slave… Given its critical importance in survival and its authoritarian command over the rest of the brain, fear should be one of the most extensively studied topics in neuroscience" (Mobbs et al. Nature Neurosci 2019). In that same wide-ranging paper, psychiatrist Kerry Ressler reminds us that, "Disorders of fear processing…are among the most common of psychiatric maladies, affecting hundreds of millions of people worldwide. Combined, they are also among the highest in terms of morbidity, loss of work, comorbid psychiatric and medical disorders, and mortality from suicide." Yet the neural systems that control the expression and regulation of fear remain incompletely understood. Considerable progress has been made using animal models, but the relevance of these tantalizing discoveries to the complexities of the human brain and human suffering remain unclear. Human fear conditioning studies represent a crucial means of the addressing this question, yet as Lonsdorf and colleagues make clear in this submission, the inferential impact of such studies is limited by important (but seemingly mundane) methodological issues. In particular, they highlight substantial heterogeneity in exclusion criteria for such studies-who does and who does not adequately learn the association between cues (CS+) and consequences (shocks; UCS). Based on this, they propose evidence-based clear guidelines for conducting and reporting fear conditioning studies.

All 3 reviewers were very enthusiastic about this submission.

• The manuscript has multiple strengths, in my opinion, including the clear writing; the focus on an obvious (to many but not all) problem in a commonly used and published paradigm; the authors' preregistration of analysis; careful and clear presentation of data; and potentially useful suggestions for improvement for the issue at hand. I believe that papers such as this one are long overdue for many subfields. Given the recently increased focus on the credibility of psychological science, this minor is especially timely.

• This is an important and consequential piece of work. The authors provide an extremely important service by surveying the current state of the literature and using an empirical approach to identify potential pitfalls with the current lack of standards and transparency. Fear conditioning is a very commonly used technique. Thus, core decisions, such as which participants to include or not include has the potential to have substantial impact on this literature, and thus for theories of learning and memory, their application to clinical problems such as anxiety and PTSD, and potentially development and improvement of interventions based on conditioning models. The topic of this paper is of clear significance to the field. In addition to significance I also found the approach used by the authors to be rigorous. The sample of papers included was sizeable and systematic. The criteria used in each paper were clearly detailed. The datasets they used to demonstrate the impact of these different exclusion criteria included large samples, minimizing spurious effects. These analyses included many useful contributions, including the dramatic loss of sample with some of the exclusion criteria, how some learning criteria are particularly impactful with respect to individual variation in magnitude of SCR response and trait anxiety (as examples), and the lack of convergence with other measures of learning (e.g., ratings). The data demonstrating the particular utility of identifying non-responders based on the US, but not the CSs, were clearly presented and compelling. The table of recommendations (Box 1) is for the most part extremely comprehensive and useful guidance for scientists in designing studies and analyzing and reporting data and methods. The suggestions for transparency in reporting and not basing identification of non-response on CSs are notably important.

• This is clearly an important topic in a field with little consensus regarding best practices around inclusion/exclusion criteria, experimental design, analysis strategies, etc. They did a good job of identifying and operationally defining the problem under study, and summarizing the literature around the problem. This combined with the thorough re-analysis of their existing data make for a compelling read and convincing case for reform.

The reviewers also outlined some important suggestions for strengthening the report, all of which can be addressed in a revision with a reasonable amount of effort by the authors.

0) The authors should strongly consider editing the title to something shorter and more direct. At present, it reads like 2-3 ideas "mashed" together.

1) The analysis of SCR discrimination as a function of "non-learning" criteria (subsection “Statistical analysis”, subsection “Applying the identified performance-based exclusion criteria to existing data: a case example”, last paragraph, Supplementary Table 3). As I understand it, the authors stratify subjects based on their CS+>CS- difference score, with lower or negative difference scores signifying non-learners. They then calculate for each group in this stratification whether or not the individuals show evidence of learning by looking at differential SCRs and fear ratings. From this, they conclude "Fourth, we provided empirical evidence that those classified as 'non-learners' (sometimes referred to as 'outliers') in SCRs based on the identified definitions ironically displayed significant CS+/CS- discrimination." In the case of differential SCRs, this argument is circular, meaning that the group definitions are dependent upon the outcome measure. Accordingly, exclusion groups will by definition differ from one another in terms of differential SCRs. In addition, exclusion groups with a lower bound > 0 will by definition show significant differential conditioning using SCRs as an outcome. This is problematic, and should be removed from the manuscript entirely. In addition, any conclusions based on this analysis should be rewritten to focus specifically on the data from the fear ratings.

2) In terms of significance for the field, I would specifically like to highlight the author's focus on characterizing the non-learners. While a small group (ideally) these individuals are usually removed from analyses and thus are not well-understood. Thus, as a field we have little understanding of the nature of non-learners. It is very possible that we are missing out on an important opportunity to understand why some individuals don't learn and what the impact of that is, for better or for worse. As an example, what does it mean of fear extinction and generalization of conditioned fear is evident in those who "didn't learn?" This may be a methodological issue, but potentially also has meaningful conceptual implications.

3) In Box 1, Section B, under "Recommendations on how to proceed" for "if 'non-learning' is determined by responding during fear acquisition training, which trial types and number of trials per trial type were considered?" the authors state "classification in SCR 'non-learners' should be based on differential scores (CS+ vs. CS-)." This leaves a lot of wiggle room. Is it possible to be more specific? This still leaves a lot of room for decisions that may result in a biased sample (such as those anxious individuals who also show pretty large responses to the CS-). And what about the number of trials? It seems that at minimal a justification of these choices should be included.

4) The authors remark in a few places that SCR is not, and should not be, the only marker of learning. This begs the question of which dependent variables should be used, how they should be treated, and how they should (or could) be combined. I realize the paper is already long, however, this could benefit from some additional brief commentary. How might researchers choose amongst dependent variables and what might they be optimal for. Would they be weighted equally? For example, some variables may be more relevant for assessing implicit vs. explicit learning. Other variables may introduce new wrinkles – for example fear potentiated startle relies on introducing another aversive stimulus (e.g., noise burst, air puff) in addition to the US. How might this impact learning? Especially fear learning?

5) I am concerned that the many important messages in this manuscript may get lost to many less careful readers due to the wealth of information presented. On one hand, I applaud the authors for being transparent and forthcoming, but I wonder if there are ways to further focus the manuscript. I believe reduction in the overall length (perhaps partly through another careful round of editing to avoid any unnecessary redundancy) would strengthen the impact of the manuscript.

6) One relevant issue not addressed in the manuscript is the rampant inconsistency of exclusionary criteria WITHIN a given lab or set of researchers. If the authors encountered this in their search, it may be worth adding a note about it. If not, I wonder if it would be worth speculating about the meaning of a given researcher repeatedly uses the same paradigm with similar samples multiple times while changing the exclusionary criteria with no justification. This could at least be addressed in the suggestions for improvements in common practices.

7) Given the proliferation of "turn key" systems and rapid adoption of affordable, (e.g., mobile) devices for measuring of psychophysiological signals, many of which have not been sufficiently vetted for their reliability, I believe a slightly greater emphasis (not necessarily much more words but perhaps stronger language)on the potential differences in exclusion of participants based on specific amplifier systems used is warranted. I would predict that some modern systems with their decreased sampling rates and other features might do a poorer job of registering smaller responses and thus might lead to more exclusions if using criteria adopted when using more robust systems. Perhaps this is worth its own study and such could be suggested as a future direction of this work?

8) Given research on differences in skin conductance responses based on sensory domain (e.g., visual versus tactile) I am not sure I agree with the authors' recommendation of using responses to the US as exclusionary criteria in these paradigms. In other words, a given participant may reliability respond to tactile stimuli but not visual stimuli.

9) Regarding recommendations and future directions: there have been calls by some researchers (e.g., https://www.ncbi.nlm.nih.gov/pubmed/12481758) and answers by others (e.g., https://www.ncbi.nlm.nih.gov/pmc/articles/PMC4715694/ ) to use manipulations and tests specifically designed to empirically demonstrate proper quantification of measures such as startle. I wonder if something similar would be helpful to determine the proper exclusionary criteria for SCR responses. For example, a manipulation designed vary potential fear conditioning learning (e.g., lesser or greater reinforcement) could be covaried with a manipulation designed to increase or decrease general skin conductance response (e.g., pharmacological or temperature variations). If the authors believe such a discussion of this is beyond the scope of the current manuscript, I would differ to their judgment.

10) I believe it would be helpful for the authors to briefly define "non-responder" on its first use in the paper (Introduction) as this term may be confused with the aforementioned "non-learners" in the same paragraph.

11) Over 400 papers were excluded for having an "irrelevant topic". I may have missed it, but I did not see any examples given of what an "irrelevant topic" was. Knowing this may help readers understand the exclusions.

12) Some spots in the supplementary materials would benefit from just a bit of text to explain the figures, rather than referring back to the main paper (for example Appendix 4—table 2).

13) It's worth considering moving Supplementary Figure 3 to the main text. While this figure is depicting just one of many decisions that may impact subject exclusion, it's a commonly used method, and the graphic depiction of the impact of the heterogeneity of these choices is quite powerful.

14) Because of their import I think it would be helpful to (briefly) more thoroughly describe the ways in which the criteria for non-learners and non-responders (Results) varied across studies. For example, it would be easier to follow if the authors explained a bit more what was meant by "Definitions differed in 1) the experimental (sub-)phases…" and so on.

15) The authors state that this work contributes to "raising awareness to the problems of performance-based exclusion of participants ('non-learners')." This seems a bit too general. The problem isn't necessarily the use of performance-based exclusion, but how that exclusion is done.

16) The authors state, "We have empirically illustrated that most definitions of 'non-learners' fail this manipulation check." Given the large number of findings and points in the manuscript, it would be helpful in the Discussion if statements like this included more detail and context to remind the reader of the finding.

17) In Box 1, General Reporting, under "recommendations for how to proceed" for "minimal response criterion (μS) to define a valid SCR," it would help to have more detail about the means by which an empirical cutoff can be determined.

[Editors' note: further revisions were requested prior to acceptance, as described below.]

Thank you for resubmitting your work entitled "Navigating the garden of forking paths for data exclusions in fear conditioning research" for further consideration by *eLife*. Your revised article has been evaluated by Floris de Lange as the Senior Editor and Alex Shackman as the Reviewing Editor.

I am very happy to accept your paper for revision pending receipt of a revision that addresses, however you see fit, one residual suggestion from the reviewers:

"My main concern in the initial version of the manuscript was the differential SCRs to stratify the sample into exclusion groups, which then are used in an analysis where differential SCRs are the dependent measure (See point 1 in revision), and the conclusion that the authors "provided empirical evidence that those classified as 'non-learners' in SCRs in the literature (sometimes referred to as 'outliers') in SCRs based on the identified definitions ironically displayed significant CS+/CS- discrimination." Although the authors added a caveat to the methods, I find that the conclusions drawn in the sixth paragraph of the subsection “‘Non-learners’: conclusions, caveats and considerations” should be based on a stronger statistical framework. My major point is that it is statistically impossible to not find statistically significant SCR differentiation in an exclusion group who's CS+ > CS- lower bound is > 0. That being said, I understand that the field has used arbitrarily defined cutoffs that could lead to some learners misclassified. I just wonder if there is a statistically stronger approach to make this point. To be clear, I think this is an important piece of work with an impactful message that I would sincerely like to endorse for publication. However, I also understand that this message could be controversial to some, and I would like the authors to present this work in such a way that it can stand up to scrutiny from those that may disagree with the message.

My recommendation would be for the authors to dramatically soften the conclusions in the aforementioned subsection and acknowledge this limitation of their approach in the Discussion section.

Alternatively, if the authors had another method for estimating the number of subjects that each cutoff criteria likely misclassifies (e.g. perhaps by using the error variance of the entire sample as a stand-in for single-subject confidence intervals), these data could be used to support the conclusions mentioned above."

---

## [Author Response]

The reviewers also outlined some important suggestions for strengthening the report, all of which can be addressed in a revision with a reasonable amount of effort by the authors.0) The authors should strongly consider editing the title to something shorter and more direct. At present, it reads like 2-3 ideas "mashed" together.

We have changed the title to “Navigating the garden of forking paths for data exclusions in fear conditioning research”.

1) The analysis of SCR discrimination as a function of "non-learning" criteria (subsection “Statistical analysis”, subsection “Applying the identified performance-based exclusion criteria to existing data: a case example”, last paragraph, Supplementary Table 3). As I understand it, the authors stratify subjects based on their CS+>CS- difference score, with lower or negative difference scores signifying non-learners.

We thank the reviewers for highlighting that this issue was not clear. We grouped subjects based on *all* the different cutoffs in SCR CS+/CS- discrimination (listed in Appendix 2—table 1, section A in the revised version) as identified from the literature search. More precisely, we did *not* classify individuals with a discrimination score of < 0 µS or = 0 µS as ‘non-learners’. What we did is to classify individuals based on the cutoff criteria identified from the literature and then test if these resulting ‘exclusion groups’ showed evidence of learning (i.e., significant discrimination) or non-learning (non-significant discrimination) in our data.

They then calculate for each group in this stratification whether or not the individuals show evidence of learning by looking at differential SCRs and fear ratings. From this, they conclude "Fourth, we provided empirical evidence that those classified as 'non-learners' (sometimes referred to as 'outliers') in SCRs based on the identified definitions ironically displayed significant CS+/CS- discrimination." In the case of differential SCRs, this argument is circular, meaning that the group definitions are dependent upon the outcome measure. Accordingly, exclusion groups will by definition differ from one another in terms of differential SCRs. In addition, exclusion groups with a lower bound > 0 will by definition show significant differential conditioning using SCRs as an outcome. This is problematic, and should be removed from the manuscript entirely. In addition, any conclusions based on this analysis should be rewritten to focus specifically on the data from the fear ratings.

We agree that the different exclusion groups, which are defined by different SCR CS+/CS- discrimination cutoff scores, can be expected to differ in SCR CS+/CS- discrimination and we also agree that this argument is somewhat circular. We have thus removed part A and B of the table (results of the ANOVAs testing for differences in SCR CS+/CS- discrimination between the exclusion groups). Yet, the fact that some or even most of these “exclusion groups” did show significant CS+/CS- discrimination in SCRs is a major problem in our opinion, as these individuals are excluded from analyses as ‘non-learners’ in the literature. Hence, we think it is an important message that there is in fact strong evidence for learning in SCRs in these groups that are referred to as ‘non-learners’. ‘Non-learners’ should not demonstrate significant discrimination per definition. The exclusion groups, except for those with cutoffs < 0 µS and = 0 µS, all showed significant SCR discrimination in our data as shown in Appendix 2—table 1. Consequentially, they showed evidence of learning in SCRs. We have made this point clearer in the Materials and methods section, the caption to Appendix 2—table 1 and revised the wording of our conclusions.

“To test whether exclusion groups differ in CS+/CS- discrimination, a mixed ANOVA with CS+/CS- discrimination in SCR or fear ratings as dependent variable and the between-subjects factor ‘Exclusion group’ and the within-subject factor ‘CS-type’ was performed. Note that it is circular to test for differences in SCR CS+/CS- discrimination between groups that were selected based on different SCR CS+/CS- discrimination cutoffs in the first place. Still, it is relevant to test whether all groups classified as ‘non-learners’ in the literature indeed fail to show evidence of learning, which would be indicated by a lack of significant CS+/CS- discrimination in SCRs in this case. In essence, this is a test to evaluate whether exclusion criteria used in the literature indeed achieve what they purport to do, that is, classify a group of participants that do not show evidence of learning.”

Appendix 2—table 1: “In this case example based on data set 1 (see main manuscript), we tested whether CS+/CS- discrimination in SCRs indeed differs between the different exclusion groups as defined by the cutoffs retrieved from the literature (see Figure 2B). […] Still, this is an important manipulation check to empirically test whether those classified as ‘non-learners’ in the literature indeed do not show evidence of learning, which would be indicated by comparable SCRs to the CS+ and the CS- (i.e., no significant discrimination).”

Results: “Unsurprisingly, the exclusion group defined by a CS+/CS- discrimination cutoff < 0 µS showed inverse discrimination (CS- > CS+, not significant in raw SCRs: p = 0.117; significant in log, rc SCRs: p = 0.021). Strikingly and more importantly, with the exception of the group formed by defining ‘non-learning’ as a discrimination score of 0 µS (light green), all other exclusion groups, which have been established by defining classified as ‘non-learners’ by the different cutoffs in the literature, showed significantly larger CS+ than CS- SCR amplitudes (i.e., significant CS+/CS- discrimination; raw: all p’s < 0.0009; log, rc: all p’s < 0.0002; see Appendix 2 –Table 1 for details).”

Discussion: “Fourth, we provided empirical evidence that those classified as ‘non-learners’ in the literature (sometimes referred to as ‘outliers’) in SCRs based on the identified definitions ironically displayed significant CS+/CS- discrimination – with the exception of non-learners defined by a cut-off in differential responding of < 0 and = 0 µS. Hence, in addition to the many conceptual problems we raised here, the operationalization of ‘non-learning’ in the field failed the its critical manipulation check, given that those classified as ‘non-learners’ show clear evidence of learning as a group (i.e., CS+/CS- discrimination, see Appendix 2—table 1).”

2) In terms of significance for the field, I would specifically like to highlight the author's focus on characterizing the non-learners. While a small group (ideally) these individuals are usually removed from analyses and thus are not well-understood. Thus, as a field we have little understanding of the nature of non-learners. It is very possible that we are missing out on an important opportunity to understand why some individuals don't learn and what the impact of that is, for better or for worse. As an example, what does it mean of fear extinction and generalization of conditioned fear is evident in those who "didn't learn?" This may be a methodological issue, but potentially also has meaningful conceptual implications.

We thank the reviewers for highlighting this important point, which we have now included in our Discussion. Yet, we would like to highlight that non-discrimination based on a single read-out measure (such as SCRs) cannot be taken as evidence for non-learning. We, however, agree that characterizing individuals that do not discriminate may be highly relevant.

Discussion: “Furthermore, by excluding these individuals from further analyses, we may miss the opportunity to understand why some individuals do not show discrimination between the CS+ and the CS- in SCRs (or other outcome measures) or whether this lack of discrimination is maintained across subsequent experimental phases. It can be speculated that this lack of discrimination may carry meaningful information – at least for a subsample.”

3) In Box 1, Section B, under "Recommendations on how to proceed" for "if 'non-learning' is determined by responding during fear acquisition training, which trial types and number of trials per trial type were considered?" the authors state "classification in SCR 'non-learners' should be based on differential scores (CS+ vs. CS-)." This leaves a lot of wiggle room. Is it possible to be more specific? This still leaves a lot of room for decisions that may result in a biased sample (such as those anxious individuals who also show pretty large responses to the CS-). And what about the number of trials? It seems that at minimal a justification of these choices should be included.

We thank the reviewers for pointing at this issue and have provided more details as suggested. Since it is very difficult to provide a universally applicable recommendation on the number of trials to be included, we follow the suggestion to provide a justification of the choices.

Box 1, section B: “classification as SCR ‘non-learners’ should be based on differential scores (CS+ vs. CS-) and the number of trials included for this calculation should be clearly justified. Providing a generally valid recommendation regarding the number of trials to be included is difficult, since it critically depends on experimental design choices.”

4) The authors remark in a few places that SCR is not, and should not be, the only marker of learning. This begs the question of which dependent variables should be used, how they should be treated, and how they should (or could) be combined. I realize the paper is already long, however, this could benefit from some additional brief commentary. How might researchers choose amongst dependent variables and what might they be optimal for. Would they be weighted equally? For example, some variables may be more relevant for assessing implicit vs. explicit learning. Other variables may introduce new wrinkles – for example fear potentiated startle relies on introducing another aversive stimulus (e.g., noise burst, air puff) in addition to the US. How might this impact learning? Especially fear learning?

We absolutely agree with the reviewers that it would be highly desirable to present a universally valid gold standard of how to quantify learning and the reviewer provides some interesting suggestions. We believe that this is an empirical question that future research needs to address. We have included these thoughts in the Discussion of our revised manuscript.

Discussion: “Furthermore, future work should empirically address the question of how to best define ‘non-learning’ in particular in light of different outcome measures in fear conditioning studies, which capture different aspects of defensive responding (Lonsdorf et al., 2017; Spielberger, Gorsuch and Lushene, 1983)”

With respect to the potential impact of startle on the fear learning process, in the revised manuscript, we now refer to our previous publication (Sjouwerman et al., 2016) that addressed this question empirically. Indeed, the reviewer is correct in assuming that inclusion of startle as an outcome measure (which involves the presentation of an aversive stimulus to trigger the startle response) slows down fear learning.

Discussion: “Procedural differences, in particular the inclusion of outcome measures that require certain triggers to elicit a response (such as startle responses or ratings), have also been shown to impact on the learning process itself (Sjouwerman et al., 2016).”

5) I am concerned that the many important messages in this manuscript may get lost to many less careful readers due to the wealth of information presented. On one hand, I applaud the authors for being transparent and forthcoming, but I wonder if there are ways to further focus the manuscript. I believe reduction in the overall length (perhaps partly through another careful round of editing to avoid any unnecessary redundancy) would strengthen the impact of the manuscript.

We thank the reviewer for bringing forward this concern and have re-read our work carefully. We have shortened and focused the content wherever we saw fit. In addition, we inserted sub-headings to the Discussion to facilitate comprehensibility and guide the readers to the points most relevant to their interests:

“‘Non-learners’: conclusions, caveats and considerations”;

“‘Non-responders’: conclusions, caveats and considerations”;

“Where do we go from here?”;

“Final remarks”.

6) One relevant issue not addressed in the manuscript is the rampant inconsistency of exclusionary criteria within a given lab or set of researchers. If the authors encountered this in their search, it may be worth adding a note about it. If not, I wonder if it would be worth speculating about the meaning of a given researcher repeatedly uses the same paradigm with similar samples multiple times while changing the exclusionary criteria with no justification. This could at least be addressed in the suggestions for improvements in common practices.

The reviewers are absolutely correct to note that there is inconsistent use of criteria not only *between* but also *within* research groups and that typically no explicit justification for this is provided. While we are well aware of this, we have deliberately refrained from pointing this out explicitly in our manuscript. We see the major contribution of our work by providing a *constructive* and *evidence-based* discussion. Identifying specific individuals or groups for using inconsistent criteria across different publications (and speculating about the underlying reasons) would, in our view, not add anything to the message we want to be heard. We have, however, followed the reviewer’s recommendation to include a note about this in our suggestions. (See Box 1, section B).

7) Given the proliferation of "turn key" systems and rapid adoption of affordable, (e.g., mobile) devices for measuring of psychophysiological signals, many of which have not been sufficiently vetted for their reliability, I believe a slightly greater emphasis (not necessarily much more words but perhaps stronger language)on the potential differences in exclusion of participants based on specific amplifier systems used is warranted. I would predict that some modern systems with their decreased sampling rates and other features might do a poorer job of registering smaller responses and thus might lead to more exclusions if using criteria adopted when using more robust systems. Perhaps this is worth its own study and such could be suggested as a future direction of this work?

We agree with the reviewers that this is a relevant implication for our work and included this in our revised Discussion.

“This being said, we do acknowledge that certain research questions or the use of different recording equipment (robust lab equipment vs. novel mobile devices such as smartwatches) may potentially require distinct data processing pipelines and potentially also exclusion of certain observations (Simmons, Nelson and Simonsohn, 2011; Silberzahn et al., 2018), hence it is not desirable to propose rigid and fixed rules for generic adoption.”

“It is difficult to suggest a universally valid cutoff with respect to the number or percentage of required valid US responses as this critically depends on a number of variables such as hardware and sampling rate used. It remains an open question for future work if data quality of novel mobile devices (e.g., smartwatches) for the acquisition of SCRs differs from traditional, robust, lab-based recordings and how this would impact on the frequency of exclusions based on SCRs.”

8) Given research on differences in skin conductance responses based on sensory domain (e.g., visual versus tactile) I am not sure I agree with the authors' recommendation of using responses to the US as exclusionary criteria in these paradigms. In other words, a given participant may reliability respond to tactile stimuli but not visual stimuli.

We agree that different sensory domains may induce SCRs of different amplitudes (e.g., Etzi and Gallace, 2016; ). In the latter study, tactile stimuli indeed induced larger SCRs with larger amplitudes compared to visual stimuli. Yet, we are not aware of empirical work showing that individuals that reliably respond to tactile stimuli do not reliably respond to visual stimuli or vice versa and hence, this would need to be tested empirically as it seems to be a hypothetical assumption as it stands now.

In our manuscript, we argue that individuals cannot be classified as ‘non-responders’ if they show SCRs to any stimulus – irrespective of stimulus modality or sensory domain. In fear conditioning experiments, the tactile US is the strongest stimulus used. We argue that individuals can be classified as ‘non-responders’ if they do not reliably respond to this strongest stimulus – which reliably elicits SCRs with much larger amplitudes and higher response frequencies than the visual CSs. As there is no empirical work on this topic, we have included Appendix 4—table 1 and 2 which illustrate non-responding rates across different stimuli (CS+, CS-, US) across twenty different experiments. These data illustrate that we can rather reliably infer ‘non-responding’ to the CSs from ‘non-responding’ to the US but not vice versa. We believe that given the above outlined arguments, our recommendation is empirically based. We added further explanations to the Discussion:

“In sum, while excluding physiological ‘non-responders’ makes sense (in terms of a manipulation check and independent of the hypothesis), we consider defining ‘non-responders’ based on the absence of SCRs to the CS as problematic (dependent on the hypothesis).”

9) Regarding recommendations and future directions: there have been calls by some researchers (e.g., https://www.ncbi.nlm.nih.gov/pubmed/12481758) and answers by others (e.g., https://www.ncbi.nlm.nih.gov/pmc/articles/PMC4715694/ ) to use manipulations and tests specifically designed to empirically demonstrate proper quantification of measures such as startle. I wonder if something similar would be helpful to determine the proper exclusionary criteria for SCR responses. For example, a manipulation designed vary potential fear conditioning learning (e.g., lesser or greater reinforcement) could be covaried with a manipulation designed to increase or decrease general skin conductance response (e.g., pharmacological or temperature variations). If the authors believe such a discussion of this is beyond the scope of the current manuscript, I would differ to their judgment.

We thank the reviewers for bringing this work to our attention. We have included a discussion on this in our revised manuscript as a suggestion for future studies.

Discussion: “Sixth, as illustrated by our case example (Figure 3), high CS+/CS- discrimination cutoffs generally favor individuals with high SCR amplitudes despite potentially identical ratios between CS+ and CS- amplitudes, which may introduce a sampling bias for individuals characterized by high arousal levels likely linked to biological underpinnings. Relatedly, future studies need to empirically address which criteria for SCR transformation and exclusions are more or less sensitive to baseline differences (for an example from startle responding see Bradford et al., 2015; Grillon et al., 2002).”

10) I believe it would be helpful for the authors to briefly define "non-responder" on its first use in the paper (Introduction) as this term may be confused with the aforementioned "non-learners" in the same paragraph.

We agree that it might be helpful to define ‘non-responders’ in the beginning of the manuscript – yet it is quite difficult, since the definitions differ across studies as outlined in our manuscript. Thus, we have included a definition in our revised manuscript that is somewhat vague but can be used as an umbrella term for the definitions found in the literature.

“Therefore, participants are often (routinely) excluded from analyses if they appear to not have not learned (‘non-learners’) or not have responded been responsive to the experimental stimuli (‘non-responders’) during fear acquisition training”.

11) Over 400 papers were excluded for having an "irrelevant topic". I may have missed it, but I did not see any examples given of what an "irrelevant topic" was. Knowing this may help readers understand the exclusions.

We thank the reviewer for highlighting this topic. We have now provided the relevant details in the revised manuscript and on OSF.

Legend of Figure 1: “Examples of irrelevant topics included studies that did not use fear conditioning paradigms (see https://osf.io/uxdhk/ for a documentation of excluded publications).”

12) Some spots in the supplementary materials would benefit from just a bit of text to explain the figures, rather than referring back to the main paper (for example Appendix 4—table 2).

We thank the reviewer for highlighting this issue and have followed the recommendation to provide more details in the appendix itself. Please note however, that the journal asked us to make substantial changes as it does not accept supplements. As such, the supplement is now split into 4 different appendices and figure supplements to main figures according to journal style.

Appendix 2—table 1: Results for 1.1 Applying the identified performance-based exclusion criteria to existing data: a case example

“In this case example based on data set 1 (see main manuscript), we tested whether CS+/CS- discrimination in SCRs indeed differs between the different exclusion groups as defined by the cutoffs retrieved from the literature (see Figure 2B). […] Still, this is an important manipulation check to empirically test whether those classified as ‘non-learners’ in the literature indeed do not show evidence of learning, which would be indicated by comparable SCRs to the CS+ and the CS- (i.e., no significant discrimination).”

Figure 3—figure supplement 1: “Title: Percentages of participants excluded (data-set 1) when employing the different CS+/CS- discrimination cutoffs (as identified by the systematic literature search and graphically shown in Figure 3B) which are illustrated as density plots in Figure 3. […] Note that the different groups are cumulative (i.e., the darker colored groups also comprise the lighter colored groups).”

Appendix 3: ***“***Results for: 1.32. Exploratory analyses on consistency of classification (‘learners’ vs. ‘non-learners’) across outcome measures and criteria employed

Throughout the main manuscript and particularly in the Discussion, we highlight that differential (CS+>CS-) SCRs alone cannot be taken to infer ‘learning’ (Figure 4—figure supplement 1).”

Figure 4—figure supplement 1

“Title: Bar plots (mean ± SE) with superimposed individual data-points showing CS+ and CS- amplitudes (raw SCR values) and CS+/CS- discrimination in (A) fear ratings and (B) SCRs raw values in the group of ‘non-learners’ as exemplarily defined for this example as a group consisting of individuals in the two lowest SCR CS+/CS- discrimination cutoff groups (i.e., ≤ 0) in data set 1. This illustrates that individuals failing to show CS+/CS- discrimination in SCRs (B), may in fact show substantial CS+/CS- discrimination (as an indicator for successful learning) in other outcome measures as exemplarily illustrated here for fear ratings (A).”

Appendix 4: Appendix 4 – Information for: Definition of ‘non-responders ‘and amount of participants excluded across studies

“In the main manuscript, we discuss different frequencies of ‘non-responding’ to different experimental stimuli (e.g., US, CS+ and CS- or in isolation or in combination) which inherently leads to different exclusion frequencies when classifying “non-responders” based on different types of stimuli. […] In addition, Appendix 4—table 2 provides information on the number and percentage of individuals in a sample showing SCR ‘non-responses’ to a certain number of US presentations during fear acquisition training as well as mean number and percentage of CS responses (CS refers to the CS+ and CS- combined) in these individuals to guide the development of empirically-based criteria to define SCR ‘non-responders’.”

13) It's worth considering moving Supplementary Figure 3 to the main text. While this figure is depicting just one of many decisions that may impact subject exclusion, it's a commonly used method, and the graphic depiction of the impact of the heterogeneity of these choices is quite powerful.

We have followed the reviewers’ suggestion and moved this figure to the main manuscript (now Figure 4).

14) Because of their import I think it would be helpful to (briefly) more thoroughly describe the ways in which the criteria for non-learners and non-responders (Results) varied across studies. For example, it would be easier to follow if the authors explained a bit more what was meant by "Definitions differed in 1) the experimental (sub-)phases…" and so on.

We have revised the respective sections accordingly:

Results: “Definitions differed in i) the experimental (sub-)phases to which they were applied (i.e., whether the full phase or only the first half, second half or even single trials were considered), ii) the number of trials that the exclusion was based on (varying between one and eight single trials), iii) the CS+/CS- discrimination cutoff applied (varying between <0 µS and < 0.1 µS) as well as iv) the CS+ type (only non-reinforced or all CS+ trials) considered for the definition. […] Of note, the number of trials included in the same ‘phase’ category is contingent on the experimental design and hence does not represent a homogeneous category (‘last half’ may include 5 trials for one study (i.e., 10 trials in total) but 10 trials for a different study employing 20 trials in total).”

Results: “The definitions differed in i) the stimulus type(s) used to define ‘non-responding’ (CS+ reinforced, CS+ unreinforced, all CS+s, CS-, US), ii) the SCR minimum amplitude criterion to define a responder (varying between 0.01 µS and 0.05 µS, or mere visual inspection), and iii) the percentage of trials for which these criteria have to be met (see Figure 6B and Appendix 1—figure 1) as well as a combination thereof.

‘Non-responding’ was most commonly defined as not showing a sufficient number of responses to the conditioned stimuli (CS+ and CS-), less frequently by the absence of responses to the US, or any stimulus (CS+, CS- or US) and in two cases by the absence of responses to the CS+ or context (CXT+) specifically (see Figure 6B).”

15) The authors state that this work contributes to "raising awareness to the problems of performance-based exclusion of participants ('non-learners')." This seems a bit too general. The problem isn't necessarily the use of performance-based exclusion, but how that exclusion is done.

We agree with the reviewers that it is not only the exclusion per se but also its implementation that is problematic and have revised this sentence accordingly:

Discussion: “Together, we believe that this work contributes to i) raising awareness to some of the problems associated with performance-based exclusion of participants (‘non-learners’) and how this is implemented […]”

16) The authors state, "We have empirically illustrated that most definitions of 'non-learners' fail this manipulation check." Given the large number of findings and points in the manuscript, it would be helpful in the Discussion if statements like this included more detail and context to remind the reader of the finding.

We thank the reviewers for pointing this out. We inserted cross-references to the relevant figures, tables and the Appendix for every discussion point to facilitate comprehensibility and to act as reminders of the respective finding. In addition, we provide more details in our revised manuscript and sub-headings as follows:

Discussion: “Fourth, we provided empirical evidence that those classified as ‘non-learners’ in SCRs in the literature (sometimes referred to as ‘outliers’) based on the identified definitions ironically displayed significant CS+/CS- discrimination – except when using cutoffs in differential responding of < 0 µS and = 0 µS. Hence, in addition to the many conceptual problems we raised here, the operationalization of ‘non-learning’ in the field failed its critical manipulation check, given that those classified as ‘non-learners’ showed clear evidence of learning (i.e., CS+/CS- discrimination, see Appendix 2—table 1).”

“‘Non-learners’: conclusions, caveats and considerations”;

“‘Non-responders’: conclusions, caveats and considerations”;

“Where do we go from here?”;

“Final remarks”.

7) In Box 1, General Reporting, under "recommendations for how to proceed" for "minimal response criterion (μS) to define a valid SCR," it would help to have more detail about the means by which an empirical cutoff can be determined.

We thank the reviewer for bringing this imprecision to our attention and have specified this in the revised manuscript:

Box 1, Section A:

“test different minimal response criteria in the data set and define the cutoff empirically. In our experience (data set 1), a cutoff was easily determined empirically by visually inspecting exemplary SCRs at different cutoffs (e.g., < 0.01 µS, between 0.01 µS and 0.02 µS) and evaluating their discrimination from noise.”

[Editors' note: further revisions were requested prior to acceptance, as described below.]

I am very happy to accept your paper for revision pending receipt of a revision that addresses, however you see fit, one residual suggestion from the reviewers:"My main concern in the initial version of the manuscript was the differential SCRs to stratify the sample into exclusion groups, which then are used in an analysis where differential SCRs are the dependent measure (See point 1 in revision), and the conclusion that the authors "provided empirical evidence that those classified as 'non-learners' in SCRs in the literature (sometimes referred to as 'outliers') in SCRs based on the identified definitions ironically displayed significant CS+/CS- discrimination." Although the authors added a caveat to the methods, I find that the conclusions drawn in the sixth paragraph of the subsection “‘Non-learners’: conclusions, caveats and considerations” should be based on a stronger statistical framework. My major point is that it is statistically impossible to not find statistically significant SCR differentiation in an exclusion group who's CS+ > CS- lower bound is > 0. That being said, I understand that the field has used arbitrarily defined cutoffs that could lead to some learners misclassified. I just wonder if there is a statistically stronger approach to make this point. To be clear, I think this is an important piece of work with an impactful message that I would sincerely like to endorse for publication. However, I also understand that this message could be controversial to some, and I would like the authors to present this work in such a way that it can stand up to scrutiny from those that may disagree with the message.My recommendation would be for the authors to dramatically soften the conclusions in the aforementioned subsection and acknowledge this limitation of their approach in the Discussion section.Alternatively, if the authors had another method for estimating the number of subjects that each cutoff criteria likely misclassifies (e.g. perhaps by using the error variance of the entire sample as a stand-in for single-subject confidence intervals), these data could be used to support the conclusions mentioned above."

We thank the reviewer for making this clearer for us and proving this feedback in a very constructive way. We agree that it is statistically very unlikely to not find a significant effect in these artificial groups – yet not impossible. We have now replaced these analyses by more conservative analyses using cumulative groups rather than within-exclusion group analyses (see Appendix 2). It is important to note that our previous conclusions still stand, as most of the cumulative exclusion groups still show significant discrimination despite of consisting of a number of individuals with negative and zero discrimination as well (in other words despite of the lower bound not being >0). With respect to the alternative suggestion of the reviewer, we consider this rather difficult as the ground truth is unknown (what is a meaningful discrimination) and hence it is unclear how to define a misclassification. We hope the reviewer and editor agree on this approach.

We have also revised the parts of the manuscript that discussed the methods of our approach or its results.

Appendix 2:Results for 1.1 Applying the identified performance-based exclusion criteria to existing data: a case example

“In this case example based on data set 1 (see main manuscript), we tested whether CS+/CS- discrimination in SCRs indeed differs between the different exclusion groups as defined by the cutoffs retrieved from the literature (see Figure 2B). Note that this is somewhat circular as exclusion groups are defined by different SCR CS+/CS- cutoffs which then are used in an analysis where differential SCRs are the dependent measure. However, that this is exactly what is sometimes done in the literature (see main manuscript).

Still, this is an important manipulation check to empirically test whether those classified as a group of ‘non-learners’ in the literature indeed do not show evidence of learning, which would be indicated by comparable SCRs to the CS+ and the CS- (i.e., no significant discrimination). Here, we test this for cumulative exclusion groups. Note that this is only a rough manipulation check as a non-significant CS+/CS- discrimination effect in the whole group (e.g., those showing a CS+/CS- discrimination <0.05µS based on raw scores) cannot be taken as evidence that all individuals in this group do not display meaningful or statistically significant CS+/CS- discrimination. More precisely, half of this group not meeting the cut-off of 0.05µS in CS+/CS- discrimination do show a negative or zero discrimination score which may bias the group average score towards non-discrimination. Yet, statistically testing for discrimination within each exclusion group (e.g. specifically in the group showing a discrimination between > 0 and < 0.05 µS) is statistically not unproblematic.”

“Strikingly and more importantly, most cumulative exclusion groups, as established by defining ‘non-learners’ by the different CS+/CS- discrimination cutoffs in SCRs in the literature, in fact show statistically significant CS+/CS- discrimination (see Appendix 2 for details and a brief discussion).”

Furthermore, we provide case-examples illustrating i) the futility of some definitions of ‘non-learners’ (i.e., when those classified as ‘non-learners’ in fact show significant discrimination on both ratings and SCRs as illustrated in Appendix 3 and Appendix 2 respectively) when applied to our data”.

“Fourth, we provided empirical evidence that those classified as a group of ‘non-learners’ in SCRs in the literature (sometimes referred to as ‘outliers’) based on the identified definitions in fact displayed significant CS+/CS- discrimination when applied to our own data. […] Hence, in addition to the many conceptual problems we raised here, the operationalization of ‘non-learning’ in the field failed its critical manipulation check given that those classified as ‘non-learners’ show clear evidence of learning as a group (i.e., CS+/CS- discrimination, see Appendix 2—table 1).”

“To test whether these exclusion groups discriminated in SCRs and fear ratings, exclusion groups were cumulated, and t-tests were performed for each cumulative group (see Appendix 2 and 3 respectively). We however acknowledge that the absence of a statistically significant CS+/CS- discrimination effect in a group on average cannot be taken to imply that all individuals in this group do not show meaningful CS+/CS- discrimination. As such, this is a rather conservative test.”